# RUFY3 links Arl8b and JIP4-Dynein complex to regulate lysosome size and positioning

Gaurav Kumar [1], Prateek Chawla[2], Neha Dhiman[2], Sanya Chadha[1], Sheetal Sharma[1], Kanupriya Sethi[1], Mahak Sharma [2] & Amit Tuli [1✉]

The bidirectional movement of lysosomes on microtubule tracks regulates their whole-cell spatial arrangement. Arl8b, a small GTP-binding (G) protein, promotes lysosome anterograde trafficking mediated by kinesin-1. Herein, we report an Arl8b effector, RUFY3, which regulates the retrograde transport of lysosomes. We show that RUFY3 interacts with the JIP4-dynein-dynactin complex and facilitates Arl8b association with the retrograde motor complex. Accordingly, RUFY3 knockdown disrupts the positioning of Arl8b-positive endosomes and reduces Arl8b colocalization with Rab7-marked late endosomal compartments. Moreover, we find that RUFY3 regulates nutrient-dependent lysosome distribution, although autophagosome-lysosome fusion and autophagic cargo degradation are not impaired upon RUFY3 depletion. Interestingly, lysosome size is significantly reduced in RUFY3 depleted cells, which could be rescued by inhibition of the lysosome reformation regulatory factor PIKFYVE. These findings suggest a model in which the perinuclear cloud arrangement of lysosomes regulates both the positioning and size of these proteolytic compartments.

[1] Divison of Cell Biology and Immunology, CSIR-Institute of Microbial Technology (IMTECH), Chandigarh, India. [2] Department of Biological Sciences, Indian Institute of Science Education and Research (IISER), Mohali, Punjab, India. ✉email: atuli@imtech.res.in

Lysosomes are heterogeneous membrane-bound organelles containing more than 60 acid hydrolases that mediate the degradation of various biological macromolecules, including proteins, carbohydrates, lipids, and nucleic acids[1]. Recent studies suggest that lysosomes are sites for storing inactive hydrolases, and the fusion of lysosomes with the cargo-containing acidic late endosomes forms a hybrid compartment-endolysosomes where most of the cargo degradation takes place[2–4]. As late endosomes, lysosomes, and endolysosomes share many commonly analyzed membrane proteins (such as LAMP1), we collectively refer to these compartments as lysosomes. Lysosomes range in numbers from 50−1000 in cultured cells and are primarily present as a relatively immobile pool in the perinuclear region of the cell (sometimes referred to as the perinuclear cloud). A minor population of lysosomes escapes from the perinuclear cloud and undergoes long-range bidirectional transport on the microtubule tracks[5,6].

Lysosomal subcellular distribution is not static and changes with the presence or absence of nutrients, growth factors, changes in cytosolic pH, exposure to oxidative stress, infection, etc.[7–15]. More importantly, cues such as nutrients and/or growth factors influence lysosome-mediated cellular responses under these physiological conditions by altering lysosomal distribution. For instance, depletion of nutrients and/or growth factors results in lysosome clustering in the perinuclear region, where the proteolytic compartments may have more propensity to tether and fuse with autophagosomes[7,16]. The degradation of autophagic cargo and subsequent recycling of breakdown products replenishes nutrient reserves under starvation conditions. In contrast, growth factor re-stimulation results in lysosome localization near the plasma membrane that facilitates reactivation of lysosomal-localized mTORC1 signaling complex, and consequently, gene expression required for protein synthesis[17]. Recent studies have also highlighted the role of lysosome positioning in promoting ER remodeling from sheets to tubules in the peripheral cellular space[18,19]. Also, the proximity of lysosomes to focal adhesions near the plasma membrane regulates lysosome-dependent focal adhesion disassembly and promotes growth factor-dependent activation of the mTORC1 signaling complex[20,21].

Several factors, including the continuous long-range motility on the microtubule tracks, association with the actin cytoskeleton, and tethering to the ER network, regulate the spatial distribution of lysosomes at the whole-cell scale. The microtubule-based motor proteins, cytoplasmic dynein in complex with dynactin and multiple kinesin family members, promote retrograde (towards microtubule minus-end) and anterograde (towards microtubule plus-end) lysosome motility, respectively[13,22]. Motor proteins are recruited to the organelle membranes by association with their adapters, generally effectors of Rab and Arf-like (Arl) small GTP-binding (G) proteins[23–25]. Rab7-RILP represents a well-characterized small G protein-effector complex that recruits the motor dynein-dynactin complex to promote retrograde motility of the late endocytic compartments[26,27]. Rab7 also interacts with FYCO1 to recruit kinesin-1 for anterograde motility of late endocytic compartments towards the plasma membrane[28].

A key player, now well known for regulating the lysosomal spatial distribution, is the small G protein Arl8[29]. Arl8 has two paralogs in mammalian cells, Arl8a and Arl8b, which are ~91% identical at the protein level and have an overlapping role in regulating lysosomal distribution. Arl8b, the better-characterized paralog, recruits its downstream effector, PLEKHM2 (also known as SKIP for SifA and Kinesin Interacting Protein) on lysosomes, which in turn recruits kinesin-1 to mediate anterograde motility of lysosomes[30,31]. Interestingly, PLEKHM1, an effector of the late endosomal small G protein Rab7, competes with SKIP/PLEKHM2 for Arl8b-binding and repositions lysosomes towards

the perinuclear region. The Arl8b-PLEKHM1 complex also promotes clustering and fusion of autophagosomes and late endosomes with lysosomes by recruiting the multisubunit tethering factor HOPS complex[32]. Arl8 paralogs also regulate KIF1A-dependent lysosome movement to the cell periphery[33].

Arl8b-mediated lysosome positioning has been shown to regulate lysosome interaction with processes occurring near the cell periphery, including growth factor-mediated activation of mTORC1, lysosome exocytosis, lysosome-mediated ER remodeling, and focal adhesion disassembly, to name a few[17,18,20,34–36]. Further, the Arl8b-SKIP complex has been shown to promote tubulation of lysosomes in activated macrophages and the formation of tubular LAMP1-positive compartments (also known as *Salmonella*-induced filaments or SIFs) in *Salmonella*-infected cells[37–39]. Recent work has also shown that Arl8b-mediated lysosomal transport to the cell periphery is required for the exit of β−coronaviruses from lysosomes, where the viruses reside before egress[40].

In addition to small G proteins and their effectors, a few studies have shown the role of lysosome membrane protein complexes in recruiting the dynein-dynactin motor, for example, MCOLN1 (TRPML1)-Alg2 and TMEM55B-JIP4 complex[9,41]. These two starvation-induced mechanisms mediate dynein-dependent transport and clustering of lysosomes in the perinuclear region. Recently, Septin9 (SEPT9), one of the Septin GTP-binding proteins, has been shown to localize to lysosomes and promote dynein-dependent retrograde transport of lysosomes[42].

Here, we report that RUN and FYVE domain-containing protein 3 (RUFY3) binds to Arl8b and recruits the JIP4-dynein-dynactin complex to Arl8b-positive lysosomes. Unlike PLEKHM1 and SKIP/PLEKHM2 (the two shared interaction partners of Arl8b and Rab7), RUFY3 did not interact with Rab7. Accordingly, upon RUFY3 depletion, there was a striking redistribution of Arl8b to the cell periphery, while Rab7 distribution was less affected. Previous studies have shown that Arl8b regulates nutrient-dependent lysosome positioning and autophagosome-lysosome fusion. RUFY3 depletion disrupted the repositioning of lysosomes to the perinuclear region in nutrient-starved cells, although the autophagic flux was not altered in these cells. Notably, endocytic cargo BODIPY-BSA cleavage was modestly reduced in RUFY3 knockdown, suggesting that lysosomes are less degradative in these cells. Along with reducing the perinuclear immobile pool of lysosomes, surprisingly, RUFY3 silencing also led to a reduction in lysosome size, which was rescued upon inhibition of lysosome reformation. Our study reveals RUFY3 as a dynein adapter that regulates the positioning of Arl8b-positive lysosomes and also impacts lysosome size, likely by regulating reformation kinetics from these compartments.

## Results

**RUFY3 is an Arl8b effector that localizes to lysosomes.** In the search for potential Arl8b-binding partners, we performed a yeast two-hybrid assay with Arl8b as bait and a human brain tissue cDNA library as prey that led to the identification of RUFY3 (NM_001037442.4; NP_001032519.1; transcript variant 1; 620 amino acids in length; longest isoform) as an interaction partner of Arl8b (Fig. 1a). Transcript variant 1 (hereafter referred to as RUFY3) is the longest transcript synthesized from the RUFY3 gene, which encodes for six alternatively spliced variants. RUFY3 variant 2 (NM_014961.5, NP_055776.1; 469 amino acids in length) is the only functionally characterized RUFY3 isoform and regulates polarity and axon growth in neurons, as well as cancer cell migration and invasion[43–49].

Using yeast two-hybrid and co-immunoprecipitation approaches, we confirmed that RUFY3 interacted with the WT (wild-type) and

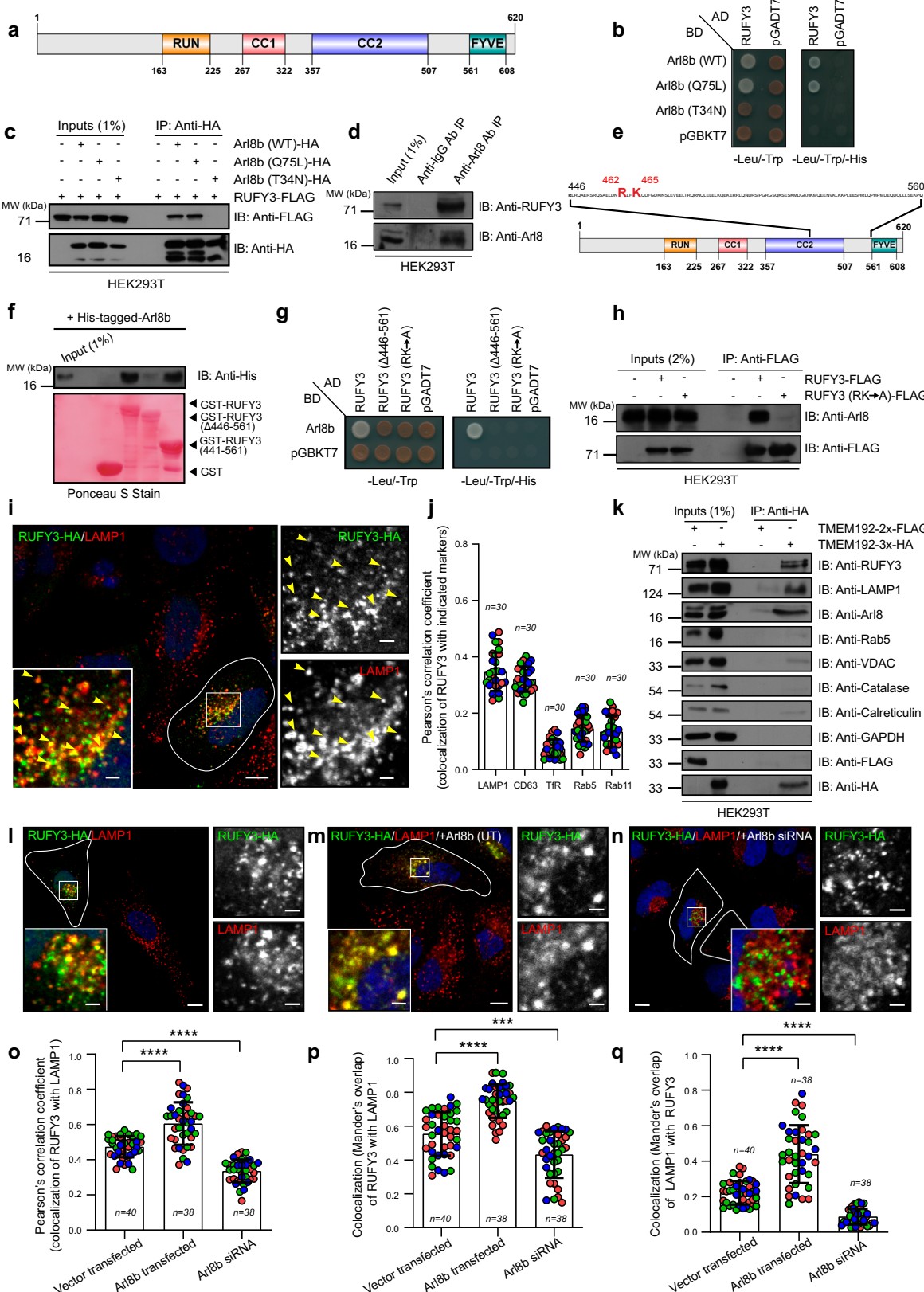

Q75L (constitutively GTP-bound) forms of Arl8b, but not with the T34N (constitutively GDP-bound) form (Fig. 1b, c). Consistent with this, RUFY3 interaction with GST-tagged-Arl8b (as bait) was reduced in the presence of excess GDP as compared to GTP, suggesting that RUFY3 behaves as an effector for the small G protein (Supplementary Fig. 1a). We also observed the interaction of Arl8b and RUFY3 under endogenous conditions by direct immunoprecipitation of Arl8b from HEK293T cell lysates (Fig. 1d).

Notably, RUFY3 variant 2 did not show an interaction with Arl8b (Supplementary Fig. 1b). Variant 1 (620 amino acids long) and variant 2 (469 amino acids long) of RUFY3 are identical in

**Fig. 1 Arl8b directly binds and recruits RUFY3 on lysosomes. a** Domain architecture of RUFY3 showing an N-terminal RUN domain, two CC (coiled-coil) domains, and a C-terminal FYVE-like domain. **b** Yeast two-hybrid assay. Cotransformants were spotted on -Leu/-Trp and -Leu/-Trp/-His media to confirm viability and interactions, respectively. **c** Lysates of HEK293T cells expressing the indicated proteins were immunoprecipitated (IP) with anti-HA antibodies-conjugated-agarose beads and immunoblotted (IB) with the indicated antibodies. **d** Endogenous IP was performed by incubating the HEK293T cell lysates with mouse anti-Arl8 antibody-conjugated-resin or mouse IgG-conjugated-resin, and IB with indicated antibodies. **e** Schematic representation of Arl8b-binding region of RUFY3 indicating the amino acid residues (R462 and K465) important for binding to Arl8b. **f** Indicated GST-tagged RUFY3 proteins immobilized on glutathione resin were incubated with purified His-Arl8b. The precipitates were IB with anti-His antibody and Ponceau S staining was done to visualize purified proteins. **g** Yeast two-hybrid assay. Cotransformants were spotted on -Leu/-Trp and -Leu/-Trp/-His media to confirm viability and interactions, respectively. **h** Lysates of HEK293T cells expressing the indicated proteins were IP with anti-FLAG antibodies-conjugated-agarose beads and IB with the indicated antibodies. **i** Confocal image of HeLa cells expressing RUFY3-HA and stained for indicated antibodies. Transfected cells are outlined and yellow arrowheads mark the localization of RUFY3 on lysosomes. **j** Colocalization of RUFY3-HA with the indicated markers was measured using Pearson's correlation coefficient (PCC). **k** Lysates were prepared from HEK293T cells expressing TMEM192-2x-FLAG (control) or TMEM192-3x-HA and subjected to LYSO-IP. The precipitates were IB with indicated antibodies. **l-n** Confocal micrographs of HeLa cells transfected with RUFY3-HA (**l**), co-transfected with RUFY3-HA and Arl8b untagged (UT) (**m**), Arl8b siRNA treated and transfected with RUFY3-HA (**n**), and stained for lysosomes using an anti-LAMP1 antibody. RUFY3 localization to LAMP1-positive compartments is shown in insets. **o-q** Colocalization of RUFY3-HA with LAMP1-positive compartments for experiments presented in **l-n** was quantified using PCC (**o**) and Mander's overlap (**p, q**). The values plotted are the mean ± S.D. from three independent experiments. Experiments are color-coded, and the total number of cells analyzed is indicated on the graph (****$p < 0.0001$; ***$p < 0.001$; two-tailed Student's $t$-test). Scale Bars: 10 μm (main); 2 μm (inset).

sequence for the first 445 amino acids. The difference between the two variants lies in a stretch of residues from 446–620, present in variant 1 but not in variant 2 (Fig. 1e and Supplementary Fig. 1c). Indeed, domain deletion analysis revealed that a RUFY3 mutant lacking residues 446–561 (hereafter referred to as RUFY3 (Δ446–561)), containing the FYVE-like domain) failed to bind to Arl8b in a yeast two-hybrid assay. More importantly, the RUFY3 fragment encompassing 441–561 residues (hereafter referred to as RUFY3 (441–561)) was sufficient for interaction with Arl8b (Supplementary Fig. 1d). This was corroborated using the GST-pulldown assay wherein Arl8b was interacting with GST-tagged-RUFY3 (WT) and -RUFY3 (441–561) but not a deletion mutant lacking these residues (Supplementary Fig. 1e). To test whether the RUFY3 fragment containing 441–561 residues directly binds to Arl8b, we incubated recombinant His-Arl8b with GST or GST-tagged-RUFY3 (WT), -RUFY3 (Δ446–561), and RUFY3 (441–561). As shown in Fig. 1f, we found that Arl8b directly binds to the RUFY3 encompassing the 441–561 fragment. The immunoblot of purified GST and GST-tagged RUFY3 proteins used in this assay is shown in Supplementary Fig. 1f.

Next, to further narrow down amino acid residues within the RUFY3 (441–561) fragment that affect binding with Arl8b, we first mutated the positively charged residues in this fragment to alanine. This selection was based on our prior study that revealed binding of effectors PLEKHM1 and SKIP/PLEKHM2 to Arl8b requires arginine residues in their RUN domain[32]. From this screening, we found that R462/K465 residues in the RUFY3 (441–561) fragment were crucial for interaction with Arl8b, as mutating these residues to alanine (RK → A) abrogated binding to Arl8b (Fig. 1e, g, h and Supplementary Fig. 1g).

We next analyzed RUFY3 localization by transfecting an epitope-tagged-RUFY3 construct into HeLa cells, as none of the available anti-RUFY3 antibodies recognized the protein under endogenous conditions. RUFY3-HA-tagged construct, when expressed in HeLa cells, showed a cytosolic distribution with few punctate structures (in <20% cells with weak to moderate level of expression) visible in the perinuclear region, which could be due to limiting expression of endogenous Arl8b (see inset, Supplementary Fig. 2a, b). To better visualize RUFY3 membrane localization that was masked by the cytosolic signal, we permeabilized the cells with a mild detergent before fixation (see inset, Supplementary Fig. 2c). Further, to elucidate the identity of the RUFY3-positive compartments, we co-stained these cells with well-characterized endosomal and lysosomal

markers. Several RUFY3 punctae were strongly colocalized with the late endosomal/lysosomal markers, LAMP1 and CD63, while little to no colocalization was observed with the early (Rab5) and recycling endosomal markers (Transferrin Receptor-TfR and Rab11) (Fig. 1i and Supplementary Fig. 2d–g; quantification is shown in Fig. 1j and Supplementary Fig. 2h, i). To corroborate whether RUFY3 localizes to lysosomes under endogenous conditions, we used the recently described LYSO-IP method that relies on immuno-purification of subcellular compartments containing the lysosomal transmembrane protein TMEM192[50]. We confirmed that the lysosomal fractions obtained using the LYSO-IP method were not contaminated with other membranes by probing for various organelle markers (Fig. 1k). RUFY3, similar to LAMP1 and Arl8b, was present in the lysosomal fractions under endogenous conditions, confirming the localization observed with the RUFY3-tagged construct (Fig. 1i and Supplementary Fig. 1j).

One of the primary roles of small G proteins of Rab, Arf, and Arl families is to recruit their effectors to target membranes; we next tested whether Arl8b plays a similar role in RUFY3 recruitment to lysosomes. Indeed, RUFY3 lysosomal localization was significantly enhanced in cells co-expressing Arl8b (see inset, Fig. 1l, m; quantification is shown in Fig. 1o–q). This increased recruitment of RUFY3 upon co-expression of Arl8b was evident from structured illumination microscopy (SIM) images of individual LAMP1-positive vesicles (compare insets of Supplementary Fig. 2j, k showing RUFY3 localization on LAMP1-positive compartments). We noted that RUFY3 localized only to a subset of LAMP1-positive compartments, even in the presence of overexpressed Arl8b (Fig. 1q, Mander's overlap of LAMP1 colocalization with RUFY3). Finally, RUFY3 recruitment to lysosomes was significantly reduced in Arl8b-depleted cells, indicating that RUFY3 behaves as an Arl8b effector (Fig. 1n; quantification shown in Fig. 1o–q). Notably, some RUFY3 punctate structures were still present in Arl8b siRNA-treated cells, but these punctae did not colocalize with LAMP1 (see inset, Fig. 1n). Whether the RUFY3 punctae in Arl8b-depleted cells represent protein aggregates or membrane-bound compartments is unclear.

**RUFY3 promotes perinuclear positioning of lysosomes.** Interestingly, lysosomes were strongly clustered in the perinuclear region upon RUFY3 transfection in HeLa cells (compare untransfected and transfected cells in Fig. 2a). To corroborate this observation, we quantified lysosomal distribution by two

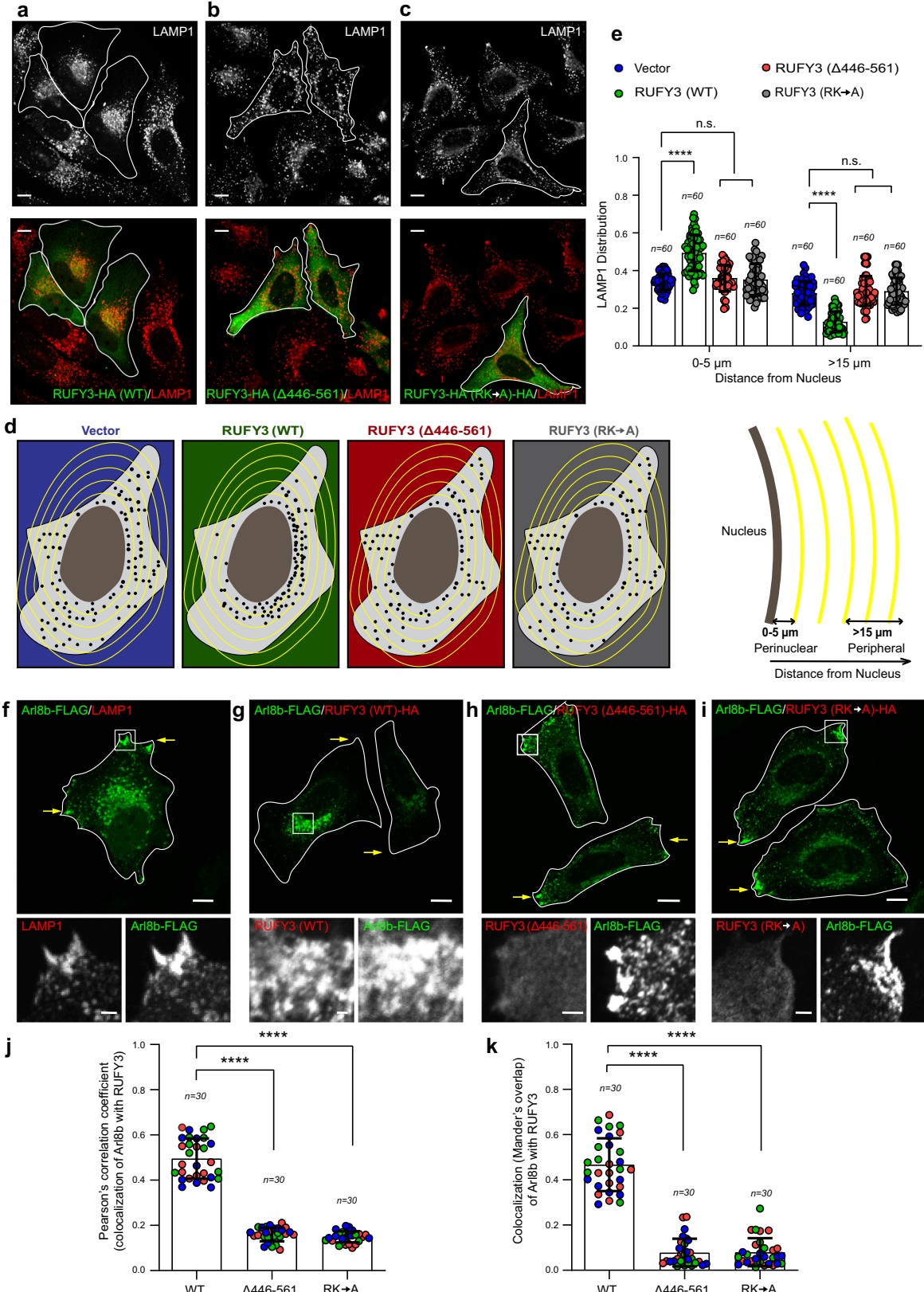

methods— (a) measuring the cumulative integrated LAMP1 intensity in the perinuclear region (0–5 μm) and the peripheral region (>15 μm) (Fig. 2d), and—(b) measuring the distance of lysosomes relative to the maximum distance from the center of the nucleus to the cell periphery[5,33,51] (Supplementary Fig. 2l) in cells transfected with either vector control or different RUFY3 constructs. As shown in Fig. 2e and Supplementary Fig. 2m, the distribution of lysosomes in RUFY3 (WT) transfected cells was significantly shifted to the perinuclear region and away from the periphery. Importantly, RUFY3 mutant proteins defective in binding to Arl8b (i.e., RUFY3 (Δ446–561) and RUFY3 (RK → A)) did not localize to the LAMP1 compartment or alter

**Fig. 2 Wild-type RUFY3, but not the Arl8b-binding-defective mutant, promotes perinuclear lysosome clustering. a–c** Confocal micrographs of HeLa cells expressing RUFY3-HA (WT) (**a**), RUFY3 (Δ446–561)-HA (**b**), and RUFY3 (RK → A)-HA (**c**) and stained for lysosomes using an anti-LAMP1 antibody. Transfected cells are marked with a boundary. **d** A schematic depicting the quantification method employed for analyzing the distribution of LAMP1-positive compartments in a cell. **e** Quantification of the distribution of LAMP1-positive compartments in HeLa cells transfected with the indicated plasmids for the experiments shown in **a–c**. The values plotted are the mean ± SD from three independent experiments. The total number of cells analyzed is indicated on the graph (****$p < 0.0001$; n.s. not significant; two-tailed Student's $t$-test). **f–i** Confocal micrographs of HeLa cells transfected with Arl8b-FLAG alone (**f**) or co-transfected with indicated RUFY3 expressing plasmids (**g–i**) and stained with indicated antibodies. The cell boundary is marked with a line and yellow arrows mark the peripheral localization of Arl8b-positive vesicles. **j, k** Colocalization analysis of Arl8b with indicated RUFY3 proteins was assessed by calculating Pearson's correlation coefficient (**j**) and Mander's overlap (**k**) from the experiments shown in **g–i**. The values plotted are the mean ± SD from three independent experiments. Experiments are color-coded, and each dot represents the individual data points from each experiment. The total number of cells analyzed is indicated on the graph (****$p < 0.0001$; two-tailed Student's $t$-test). Scale Bars: 10 µm (main); 2 µm (inset).

lysosome positioning, suggesting that association with Arl8b is required for RUFY3 lysosomal localization (Fig. 2b, c; quantification is shown in Fig. 2e and Supplementary Fig. 2m).

From several previous studies[29,30,52,53], it is known that Arl8b is enriched on peripheral lysosomes, and its overexpression drives the accumulation of lysosomes near the plasma membrane (see inset, Fig. 2f). This is attributed to Arl8b interaction with a RUN domain-containing protein, SKIP/PLEKHM2, that binds and recruits the kinesin-1 motor to drive the anterograde motility of late endosome/lysosome (LE/Lys) on microtubule tracks[31,54]. Interestingly, co-expression of RUFY3 caused a striking shift in Arl8b distribution to the perinuclear region wherein both proteins colocalized on these perinuclear compartments (see inset, Fig. 2g; Pearson's and Mander's colocalization coefficients are shown in Fig. 2j, k). Consistent with our analysis of the residues of RUFY3 required for Arl8b-binding, no significant colocalization or a change in Arl8b distribution was observed in cells expressing RUFY3 (Δ446–561) and RUFY3 (RK → A) mutants (Fig. 2h, i; Pearson's and Mander's colocalization coefficients are shown in Fig. 2j, k). Thus, our data suggest that RUFY3 is an Arl8b effector that promotes the perinuclear positioning of lysosomes.

**RUFY3 is essential and sufficient to drive perinuclear lysosome positioning.** We used two independent strategies to corroborate whether RUFY3 is essential and sufficient to drive LE/Lys perinuclear positioning. Using the RNA interference approach (siRNA and shRNA), we depleted RUFY3 in HeLa cells and analyzed lysosome distribution. The efficiency of RUFY3 silencing was found to be >90%, as confirmed by Western blotting (Supplementary Fig. 3a, b). To monitor lysosomal distribution, besides LAMP1, we also employed Lysotracker and SiR-Lysosome probes that mark acidic and degradative (specific for lysosomal protease cathepsin D) compartments, respectively. Consistent with our results that RUFY3 expression promotes perinuclear clustering of lysosomes, RUFY3 depletion had the opposite effect, i.e. lysosomes were now localized to the cell periphery (for LAMP1 distribution, see Fig. 3a, b and Supplementary Fig. 3c–e; for Lysotracker distribution, see Fig. 3c, d; for SiR-Lysosome distribution, see Fig. 3e, f). Notably, in these experiments, only a subset of the lysosomes were relocated to the periphery upon RUFY3 depletion, and a modestly reduced perinuclear pool of lysosomes was still present in RUFY3-depleted cells. Peripheral lysosomal distribution was rescued in cells expressing the siRNA-resistant RUFY3 construct, indicating that the phenotype was specifically due to RUFY3 depletion and not due to the off-target effect of siRNA oligos (Fig. 3a–f and Supplementary Fig. 3c).

RUFY3 depletion in other cell types, including ARPE-19 (retinal pigment epithelial cells), U2OS (osteosarcoma cells), and A549 (lung adenocarcinoma cells), showed a similar distribution of lysosomes towards the cell periphery (Supplementary Fig. 3f–h). Notably, we also found that RUFY3-

depleted cells had a ~1.3-fold increase in their surface area compared to control siRNA or shRNA-treated cells (Supplementary Fig. 3i, j). Interestingly, cell spreading is reduced upon Myrlysin gene knockout, where lysosomes are clustered in the perinuclear region[36]. In contrast, the surface area of cells is increased upon dynein depletion[55], where lysosomes, similar to RUFY3 depletion, are localized to the cell periphery. These observations suggest that lysosome distribution might regulate cell spreading, but the mechanistic basis of how this is achieved remains unclear.

Next, we used the knockout-sideways approach to test whether the presence of RUFY3 on the organelle membrane was sufficient to drive their positioning to the perinuclear region. To this end, we used the FRB-FKBP rapamycin-induced heterodimerization system to mislocalize RUFY3 to mitochondria (where it is not present under endogenous conditions) and analyzed mitochondria distribution (Fig. 3g). As expected, we found mitochondrial localization of FKBP-GFP (vector transfected) and FKBP-GFP-RUFY3 fusion protein in the presence of rapamycin but not in untreated cells (Fig. 3h). Notably, in the presence of rapamycin, RUFY3-transfected cells showed a dramatic clustering of mitochondria in the perinuclear region. In contrast, vector-transfected cells showed typical mitochondrial distributions (compare second and fourth panels, Fig. 3h). Quantification of mitochondrial intensity distribution showed an increased perinuclear index in cells expressing FKBP-GFP-RUFY3 in the presence of rapamycin (Fig. 3i and Supplementary Fig. 3k). Taken together, we conclude that RUFY3 localization to the organelle membrane is sufficient to drive their distribution to the perinuclear region.

**RUFY3-mediated perinuclear lysosome positioning is independent of Rab7.** We were intrigued by the observations that only a subset of LAMP1/Lysotracker/SiR-Lysosome-positive vesicles responded to RUFY3 depletion and relocated towards the cell periphery. The two small G proteins, Rab7 and Arl8b, and their downstream effectors primarily localize to and regulate the distribution of late endocytic compartments[32,51]. We, therefore, sought to investigate whether RUFY3 is a specific or shared adapter of both Arl8b and Rab7. To this end, we first determined whether RUFY3 interacts with Rab7. In a yeast two-hybrid assay, RUFY3 did not bind to Rab7 but showed interaction with Arl8b (Fig. 4a). RILP, a well-characterized Rab7 effector, was used as a positive control and expectedly showed interaction with Rab7. Supporting this result, we observed co-immunoprecipitation of Arl8b, but not of Rab7, with RUFY3 (Fig. 4b). Thus, unlike PLEKHM1 and SKIP/PLEKHM2 (the two shared interaction partners of Arl8b and Rab7)[32,51], RUFY3 did not interact with Rab7. We next investigated whether Rab7 regulates RUFY3 membrane localization using two approaches. First, in cells expressing artificial fusion constructs of Rab7 and Arl8b with a mitochondrial targeting sequence, we analyzed whether RUFY3 is

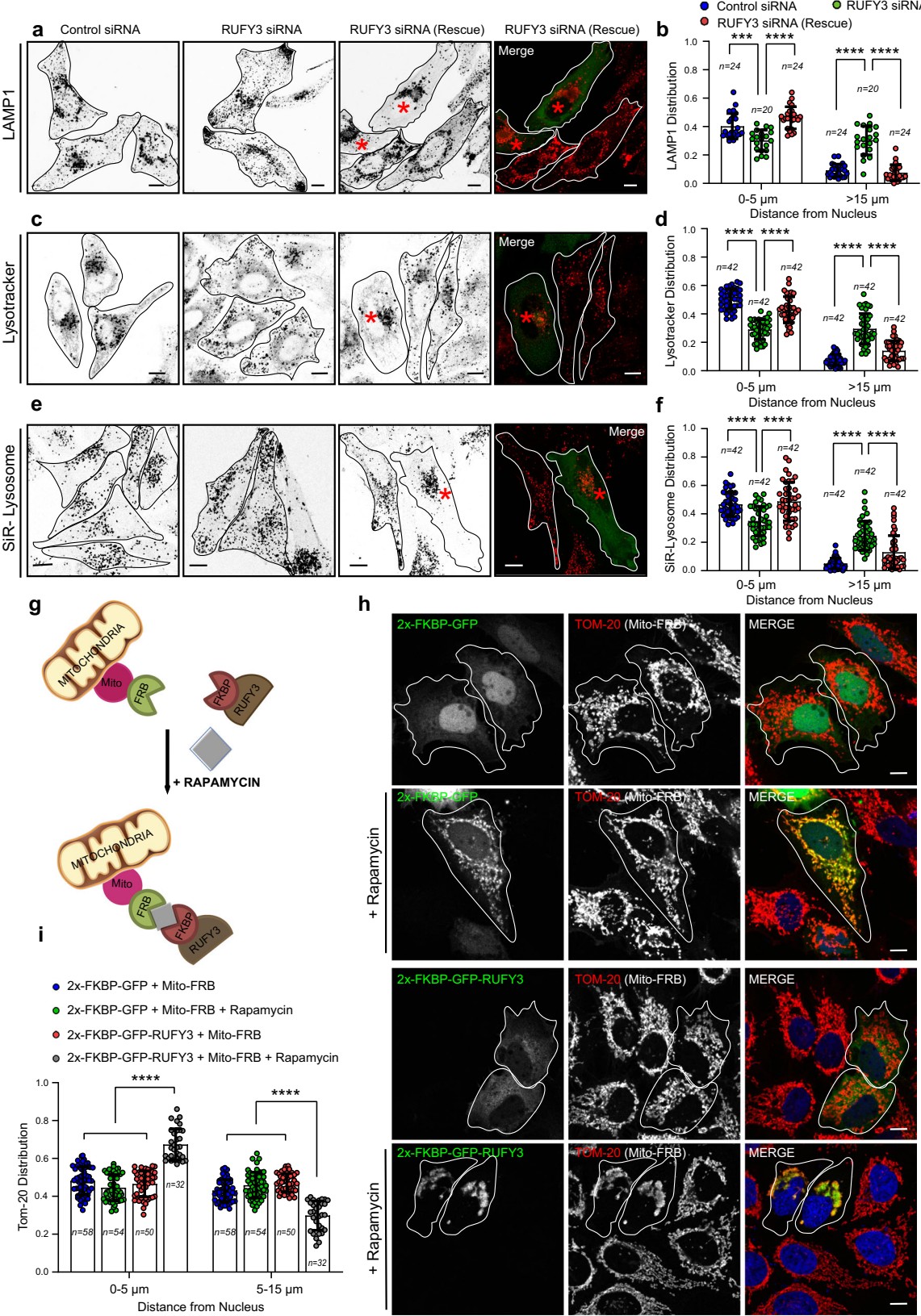

recruited to mitochondria. Consistent with our earlier results that RUFY3 interacts with Arl8b, RUFY3 was recruited to mitochondria in the presence of Mito-Arl8b but not Mito-Rab7 (Fig. 4c, d; quantification shown in Fig. 4e). Second, we analyzed whether RUFY3 localizes to and alters lysosomal distribution in Rab7-depleted cells. As shown in Fig. 4f–h, RUFY3 continued to

colocalize with LAMP1 in Rab7-depleted cells, indicating that RUFY3 membrane localization is independent of Rab7. Consistent with RUFY3 localization, RUFY3-dependent lysosome perinuclear clustering was observed in Rab7-depleted cells (Fig. 4i). Finally, we tested the impact of RUFY3 depletion on the positioning of endogenous Rab7 and Arl8b compartments in the

**Fig. 3 RUFY3 is essential and sufficient to drive perinuclear lysosome positioning. a–f** Confocal micrographs showing lysosome distribution in HeLa cells treated with the indicated siRNAs. The lysosomes were stained using an anti-LAMP1 antibody (**a**), a Lysotracker probe (**b**), or a SiR-Lysosome probe (**e**). Cells expressing the RUFY3 siRNA-rescue construct are marked by asterisks, and the image panels are shown in an inverted grayscale. The distribution of lysosomes was quantified from these experiments and shown in **b**, **d**, and **f**. The values plotted are the mean ± SD from three independent experiments. The total number of cells analyzed is indicated on the graph (****$p < 0.0001$; ***$p < 0.001$; two-tailed Student's $t$-test). **g** Schematic representation of the rapamycin-inducible FRB/FKBP protein–protein interaction. **h** Confocal micrographs of untreated- and rapamycin-treated HeLa cells expressing Mito-FRB with 2x-FKBP-GFP or 2x-FKBP-GFP-RUFY3. To visualize mitochondria, cells were stained using an anti-TOM-20 antibody, and transfected cells are marked with a white boundary. **i** The distribution of mitochondria based on the TOM-20 signal was quantified from the experiments shown in **h**. The values plotted are the mean ± SD from three independent experiments. The total number of cells analyzed is indicated on the graph (****$p < 0.0001$; two-tailed Student's $t$-test). Scale Bars: 10 μm.

same cell. Quantification of the intensity profile distribution of both Rab7 and Arl8b revealed striking peripheral relocalization of Arl8b compartments, while Rab7 distribution showed a modest but significant increase in the cell periphery (Fig. 4j, k; quantification is shown in Fig. 4l). This altered Rab7 distribution upon RUFY3 depletion is not surprising, as Rab7 and Arl8b colocalize together on a subset of late endocytic compartments (thought to be endolysosomes formed by fusion of late endosomes and lysosomes)[32,51]. Still, RUFY3 depletion affected the spatial organization of the two G proteins, as evident by a modest reduction in Rab7 and Arl8b colocalization (Fig. 4m). Taken together, these findings indicate that RUFY3 is a specific Arl8b effector that regulates the distribution of lysosomes marked by Arl8b.

**RUFY3 recruits the JIP4-dynein-dynactin complex to mediate retrograde transport of lysosomes**. To investigate the RUFY3 mode of action, we performed a GST-pulldown assay with GST-RUFY3 as a bait protein to identify potential interaction partners. Interestingly, in the GST-RUFY3 eluate, we found peptides corresponding to cytoplasmic dynein heavy chain (DYNC1H1/ DHC); dynactin 1/p150glued (DCTN1), a subunit of the dynactin complex that mediates dynein activation; and peptides of JIP4/ SPAG9 scaffolding protein that interact with dynein/dynactin and link dynein to the organelle membranes[9,56] (Supplementary Data 1). We confirmed RUFY3 and JIP4 interaction by incubating recombinant GST-RUFY3 protein with semi-purified FLAG-tagged-JIP4 isolated from mammalian cells. As shown in Fig. 5a, b, JIP4 was bound to purified GST-RUFY3 but not GST, implying JIP4 interacts with RUFY3. We also confirmed that JIP4 and RUFY3 form a complex under endogenous conditions by immunoprecipitation of both RUFY3 and JIP4 and probing for the corresponding partner. Dynein and dynactin subunits were also co-immunoprecipitated in the RUFY3-JIP4 complex (Fig. 5c–f). To test whether RUFY3 recruits JIP4 to Arl8b-positive lysosomes, we analyzed JIP4 localization in cells either expressing Arl8b alone or co-expressing both Arl8b and RUFY3. We found enhanced colocalization of Arl8b and JIP4 in the presence of RUFY3 (Fig. 5g–i). Recruitment of the p150glued dynactin subunit to Arl8b-positive structures was also increased in cells co-expressing RUFY3 (Supplementary Fig. 4a–c). In agreement with these immunofluorescence observations, immunoprecipitation data confirmed that JIP4 interaction with Arl8b was dependent upon RUFY3 expression levels (Fig. 5j–m). We next tested whether dynein and JIP4 are required for the RUFY3-mediated perinuclear clustering of lysosomes. RUFY3 overexpression failed to cause perinuclear clustering of lysosomes in JIP4- or dynein-depleted cells, suggesting that the JIP4-dynein motor complex is required for RUFY3-mediated perinuclear lysosome positioning (Fig. 5n, o and Supplementary Fig. 4d; quantification is shown in Fig. 5p). Notably, JIP4 and dynein depletion had a more profound effect than RUFY3 on lysosome distribution, with ~45% of Lysotracker-positive vesicles now localized to the cell periphery upon JIP4 and dynein depletion, as compared to ~25% in

RUFY3-depleted cells (Supplementary Fig. 4e–i). Indeed, these findings support the overall hypothesis that RUFY3 is a dynein adapter for a subset of lysosomes (primarily Arl8b-positive) and suggest the existence of other lysosomal adapters, such as TMEM55B, which binds to the JIP4-dynein-dynactin complex and mediates retrograde lysosome motility[9].

These conclusions led to a hypothesis that RUFY3 recruits the dynein motor on lysosomes and thereby mediates dynein-dependent lysosomal perinuclear positioning. Indeed, the motility behavior of lysosomes (labeled with Lysotracker) analyzed by tracking individual lysosomes showed that, similar to dynein depletion, RUFY3 depletion significantly increased the total mobile fraction and the average speed of individual lysosomes (Fig. 6a–c; see Supplementary Movies 1–3; quantification is shown in Fig. 6d, e). Thus, our data suggest that upon RUFY3 and/or dynein depletion, there is an increase in the proportion of mobile lysosomes. To directly analyze whether RUFY3 regulates dynein subunit levels on lysosomes, we used density gradient ultracentrifugation to enrich lysosomes from control and RUFY3-depleted cells. Indeed, upon RUFY3 depletion, dynein intermediate chain (DIC) levels were reduced in the lysosomal fractions compared to the control cells (Fig. 6f, g). We noted that DIC levels in other fractions were also reduced in RUFY3-depleted homogenates, suggesting that RUFY3 might regulate dynein levels on other compartments as well. Finally, based on our hypothesis, we predicted that the expression of other dynein adapters that localize to LAMP1 compartments should reinstate dynein-dependent lysosome positioning in RUFY3-depleted cells. Indeed, RILP and TMEM55B, both of which interact with and recruit dynein-dynactin on the LAMP1 compartment[9,27], repositioned lysosomes to the perinuclear region in RUFY3-depleted cells (compare untransfected with transfected cells, Fig. 6h–j; quantification is shown in Fig. 6k). Taken together, these findings show that RUFY3 is an Arl8b effector that recruits dynein on lysosomes to maintain the typical stable pool of immobile lysosomes localized in the perinuclear region of the cell.

**Depletion of RUFY3 reduces lysosome size**. Previous studies have shown that the perinuclear and peripheral pools of lysosomes have few differential characteristics and functions. The peripheral pool of lysosomes is more poised for crosstalk and fusion with the plasma membrane and serum-dependent-mTORC1 activation[7,17,35]. In contrast, the perinuclear lysosomal subpopulation is more suited for interaction with perinuclear late endosomes/autophagosomes and, subsequently, cargo degradation[16]. Moreover, in at least one study, it has been reported that the peripheral pool of lysosomes is less acidic and less accessible to biosynthetic cargo (such as cathepsins)[8]. However, a subsequent report has shown that peripheral and perinuclear lysosomes have a similar pH (~4.4)[57].

Since RUFY3 depletion results in an increased lysosomal pool near the plasma membrane, we wanted to determine whether lysosome characteristics including, their pH, size, and number,

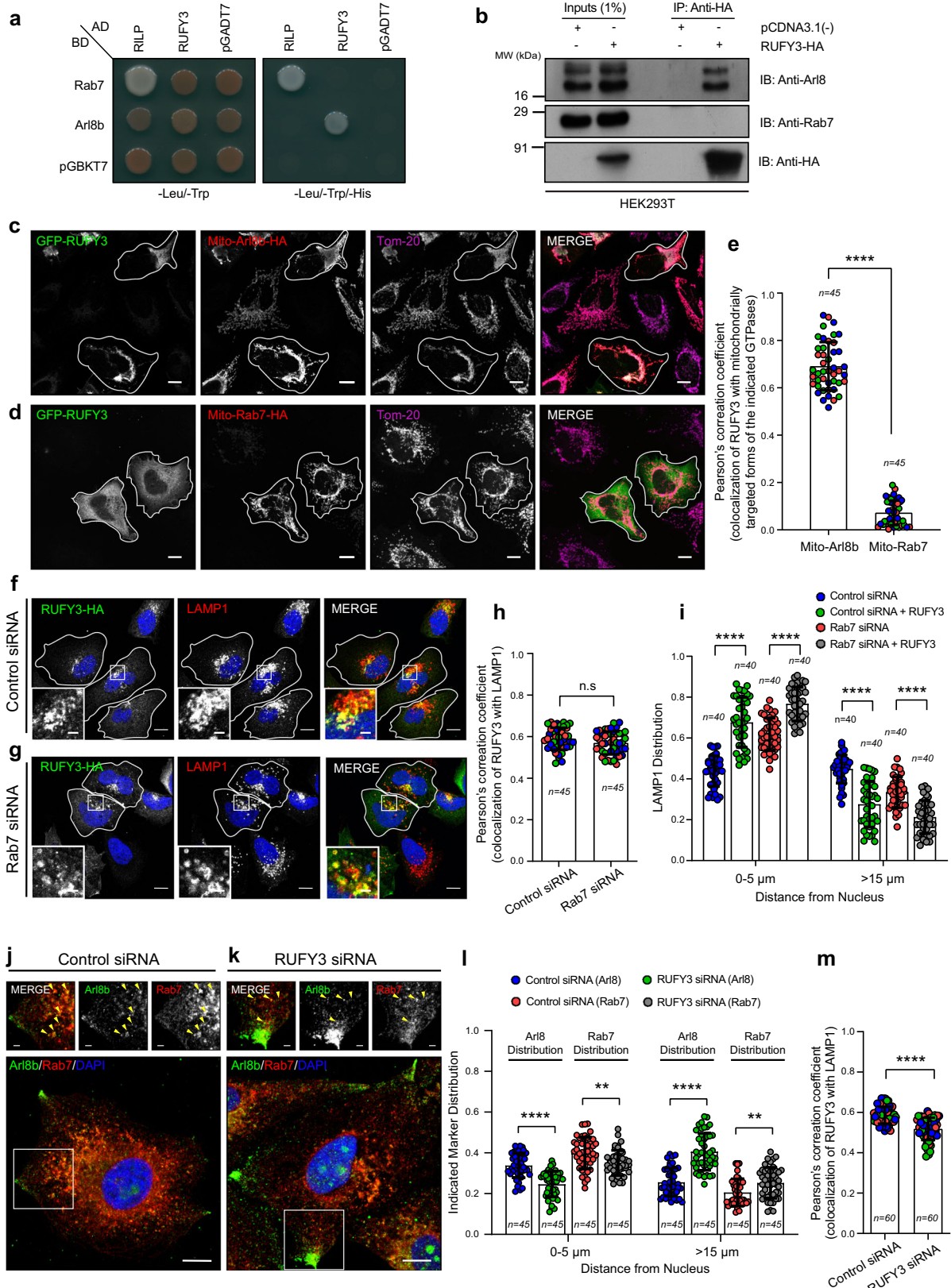

are altered in these cells. We used fluorescent dyes Lysotracker and Lysosensor Yellow/blue DND-160, which have different characteristics but share the property of fluorescing in acidic compartments[58]. Intensity variations in Lysotracker staining report on the size and number of acidic compartments but cannot report variations in pH within the acidic range[59].

Lysosensor dyes are pH sensitive and are used for ratiometric measurement of the intraorganellar pH of acidic organelles[60]. Surprisingly, while we did not observe any significant changes in lysosome pH in RUFY3-depleted cells ($5.63 \pm 0.19$), as compared to control cells ($5.49 \pm 0.18$) (Fig. 7a, b), there was a two-fold reduction in Lysotracker intensity in RUFY3-depleted cells, as

**Fig. 4 RUFY3-mediated perinuclear lysosome positioning is independent of Rab7. a** Yeast two-hybrid assay. The cotransformants were spotted on -Leu/-Trp and -Leu/-Trp/-His media to confirm viability and interactions, respectively. **b** Lysates of HEK293T cells expressing RUFY3-HA were IP with anti-HA antibodies-conjugated-agarose beads and the precipitates were IB with the indicated antibodies. **c, d** Confocal micrographs of HeLa cells co-transfected with GFP-RUFY3 and mitochondria localization tagged-Arl8b (Mito-Arl8b-HA) or -Rab7 (Mito-Rab7-HA) and stained with indicated antibodies. Transfected cells are marked with a boundary. **e** Colocalization analysis of GFP-RUFY3 with Mito-Arl8b-HA and Mito-Rab7-HA proteins was assessed by calculating Pearson's correlation coefficient for the experiments shown in **c** and **d**. The values plotted are the mean ± SD from three independent experiments. Experiments are color-coded, and each dot represents the individual data points from each experiment. The total number of cells analyzed is indicated on the graph (****$p < 0.0001$; two-tailed Student's $t$-test). **f–i** Confocal micrographs of HeLa cells treated with control siRNA (**f**) or Rab7 siRNA (**g**) and transfected with RUFY3-HA. The cells were stained for lysosomes and RUFY3 using anti-LAMP1 and anti-HA antibodies, respectively. Transfected cells are marked with a boundary. In the insets, a magnified region of the boxed area is shown indicating the localization of RUFY3 on lysosomes. Quantification of colocalization analysis of RUFY3 with LAMP1 and the distribution of lysosomes from these experiments are shown in **h**, **i**, respectively. The values plotted are the mean ± SD from three independent experiments. The total number of cells analyzed is indicated on the graph (****$p < 0.0001$; n.s. not significant; two-tailed Student's $t$-test). **j–m** Confocal micrographs of HeLa cells treated with control siRNA (**j**) or RUFY3 siRNA (**k**) and stained for endogenous Arl8 and Rab7. In the insets, the distribution of Arl8 and Rab7 is shown along with yellow arrowheads marking colocalized pixels. The distribution and colocalization of Arl8- and Rab7-positive endosomes from these experiments are shown in **l**, **m**, respectively. The values plotted are the mean ± SD from three independent experiments. The total number of cells analyzed is indicated on the graph (****$p < 0.0001$; **$p = 0.0014$ (for 0–5 μm); **$p = 0.0026$ (for >15 μm); two-tailed Student's $t$-test). Scale Bars: 10 μm (main); 2 μm (inset).

compared to control (Fig. 7c, d). The decrease in Lysotracker intensity was rescued in cells expressing a RUFY3 siRNA-resistant construct, indicating that this phenotype is specifically due to RUFY3 depletion (Fig. 7e; quantification is shown in Fig. 7f). These findings suggest that lysosome size is affected by RUFY3 depletion.

To directly assess lysosome size, we analyzed the ultrastructure of LE/Lys by transmission electron microscopy imaging on thin sections of control and RUFY3-depleted cells. As compared to control, lysosomes appeared to be smaller, denser, and more numerous upon RUFY3 depletion (see insets, Fig. 7g, h). The diameter of lysosomes was reduced by ~20% in RUFY3-depleted cells compared to control, which would translate into a ~50% reduction in lysosome volume (Fig. 7i)[61]. We noted a ~1.8-fold increase in lysosome (multi-lamellar structures) numbers in RUFY3-depleted cells compared to control cells (Fig. 7j). We corroborated these observations by measuring the average area and number of LAMP1-positive compartments from super-resolution imaging of control and RUFY3 knockdown cells. As shown in Fig. 7k–n, there was a significant reduction in the average area of lysosomes and a corresponding increase in lysosomes numbers in RUFY3-depleted cells.

The mechanism of how RUFY3 regulates lysosome size remains unclear at this time. One of the processes that could result in decreased lysosome size and increased numbers is the membrane fission of these late endocytic compartments. The enzyme PIKFYVE that forms PI(3,5)P2 from PI(3)P has been previously shown to regulate lysosome size by promoting tubulation and fission[62–64]. We investigated whether PIKFYVE depletion would restore normal lysosomal size in RUFY3-depleted cells. Indeed, cells co-depleted of RUFY3 and PIKFYVE showed a normal size distribution of lysosomes (see insets in Fig. 7o–q; quantification is shown in Fig. 7r). Thus, RUFY3 not only maintains a perinuclear lysosomal pool but also regulates lysosome size. It will be interesting to determine whether the RUFY3-JIP4-dynein complex regulates both lysosome positioning and lysosome reformation.

**RUFY3 regulates nutrient-dependent lysosome repositioning but not autophagic cargo clearance.** Previous reports have shown that Arl8b and its upstream regulator-BORC complex regulate nutrient-dependent lysosome positioning to the cell periphery[35,36]. Based on our findings that RUFY3 functions as a dynein adapter on lysosomes, we expected that RUFY3-depleted cells would fail to show repositioning of lysosomes to the perinuclear region in nutrient-starved cells. Indeed, lysosomes continued to localize at the cell periphery in RUFY3-depleted cells that were subjected to either complete starvation (EBSS-media lacking both serum factors and amino acids) or serum starvation (DMEM-FBS) or only amino acid (DMEM-AA) (Fig. 8a–d). This was in contrast to the control siRNA-treated cells, whereas expected, lysosomes were accumulated in the perinuclear region and generally absent from the periphery in all three conditions of starvation (Fig. 8a–d; quantification is shown in Fig. 8e).

Lysosome clustering to the perinuclear region in nutrient-deprived cells has been shown to result in the enhanced propensity of fusion with mature autophagosomes, which is important for replenishing the macromolecular building blocks in the starved cells[16]. The fusion of autophagosomes and lysosomes and the degradation of autophagic cargo have classically been measured by the amount of autophagosomal protein LC3B remaining in the cells with/without starvation[65]. To address the RUFY3 role in autophagic cargo degradation, we assessed the amount of lipidated LC3 (LC3B-II) levels in fed and starved cells treated with control or RUFY3 siRNA. As shown in Fig. 8f, while the initial levels of LC3B-II were modestly lower in the fed state upon RUFY3 depletion, upon EBSS treatment, both control, and RUFY3-depleted cells showed a similar increase in LC3B-II levels. Also, LC3B-II levels were rescued to a similar extent in control and RUFY3-depleted cells treated with Bafilomycin A1 (BafA1), an inhibitor of lysosomal acidification and, therefore, degradation (Fig. 8g). These results suggest that RUFY3 does not regulate autophagosome-lysosome fusion. To corroborate the autophagy flux analysis, we also measured the colocalization between LC3 and LAMP1 in serum-starved-control and -RUFY3-depleted cells treated with BafA1 to ensure the maximal frequency of autolysosomes is observed in these experiments. While there was a modest decrease in the LC3/LAMP1 colocalization in RUFY3-depleted cells, the difference in average Pearson correlation coefficient values from control was minor and not significant (Fig. 8h, i; quantification is shown in Fig. 8j). We noted that several peripheral lysosomes in RUFY3-depleted cells were also colocalized with LC3, suggesting that autolysosome formation is also occurring outside the perinuclear region (see inset in Fig. 8i). Thus, while lysosome repositioning to the perinuclear subcellular location was strikingly reduced upon RUFY3 depletion, no significant changes in autophagosome-lysosome fusion and LC3 flux were observed in RUFY3-depleted cells. Our findings agree with previous work showing that peripheral lysosomes can also undergo fusion with autophagosomes[66].

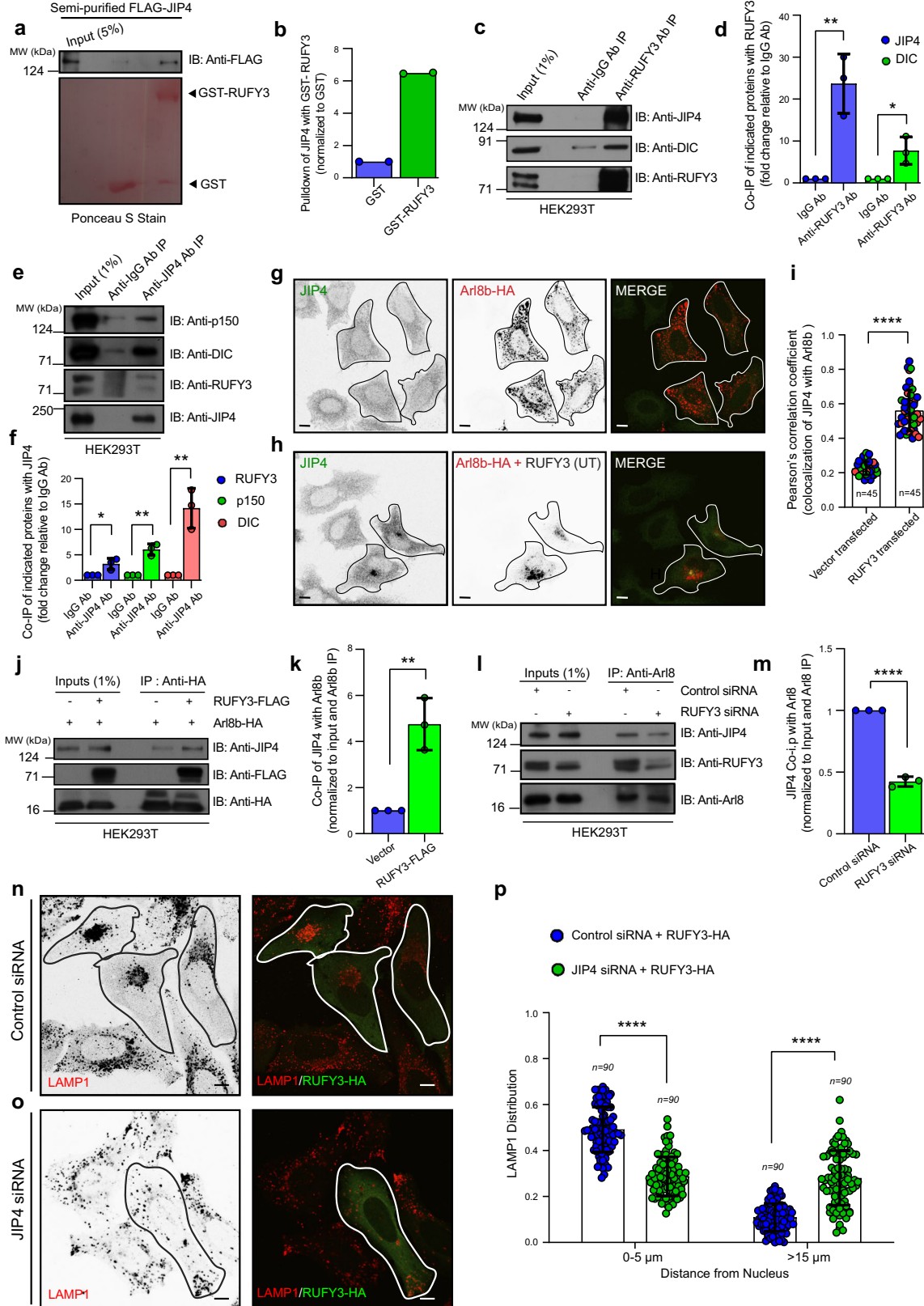

We also assessed whether RUFY3 regulates the delivery of endocytic cargo to lysosomes. To this end, we pulsed control and RUFY3-depleted cells with BODIPY-LDL followed by a chase for different time points and determined colocalization with lyso-tracker compartments. In addition, we also tested the colocalization of endocytosed dextran with Lysotracker compartments in control and RUFY3-depleted cells. As shown in Supplementary Fig. 5a–c, e–g, there was no significant change in colocalization of LDL or dextran with Lysotracker upon RUFY3 depletion (quantification shown in Supplementary Fig. 5d, h), suggesting that RUFY3 does not regulate delivery and fusion of endocytic or autophagic cargo vesicles to lysosomes. Interestingly, there was a

**Fig. 5 RUFY3 links Arl8b to the JIP4-dynein complex. a**, **b** GST-pulldown assay of semi-purified FLAG-tagged-JIP4 with GST and GST-RUFY3 and immunoblotted (IB) with anti-FLAG antibody. GST proteins were visualized by Ponceau S staining. Quantification of blots from two independent experiments is shown in **b. c**–**f** Lysates of HEK293T cells were subjected to endogenous IP as labeled and the precipitates were IB with the indicated antibodies. Quantification of the blots is shown in **d**, **f**. The values plotted are the mean ± SD from three independent experiments. For **d**, **\*\***$p = 0.0052$ and **\***$p = 0.0236$ and for **f**, **\*\***$p = 0.0015$ (p150); **\*\***$p = 0.0043$ (DIC), and **\***$p = 0.0251$ (two-tailed Student's $t$-test). **g**–**i** Confocal micrographs of HeLa cells transfected with Arl8b-HA (**g**) or co-transfected with RUFY3 (UT) (**h**) and stained with indicated antibodies. Transfected cells are outlined, and some panels are shown in an inverted grayscale. The colocalization of JIP4 with Arl8b was measured by Pearson's correlation coefficient (**i**). The values plotted are the mean ± SD from three independent experiments. Experiments are color-coded, and each dot represents the individual data points from each experiment. The total number of cells analyzed is indicated on the graph (**\*\*\*\***$p < 0.0001$; two-tailed Student's $t$-test). **j**, **k** HEK293T cell lysates expressing Arl8b-HA or co-expressing Arl8b-HA and RUFY3-FLAG were subjected to IP and the precipitates were IB with the indicated antibodies. Quantification of the blot is shown in **k** and values plotted are mean ± SD from three independent experiments (**\*\***$p = 0.0045$; two-tailed Student's $t$-test). **l**, **m** HEK293T cells were treated with the indicated siRNAs and subjected to endogenous IP using an anti-Arl8 antibody. The precipitates were IB with the indicated antibodies. The quantification of the blot is shown in **m** and values plotted are the mean ± SD from three independent experiments (**\*\*\*\***$p < 0.0001$; two-tailed Student's $t$-test). **n**, **o** Confocal images of HeLa cells treated with indicated siRNAs and transfected with RUFY3-HA. The cells were stained with anti-LAMP1 and anti-HA antibodies, respectively. Transfected cells are outlined, and some panels are shown in an inverted grayscale. **p** The distribution of lysosomes was quantified from the experiments shown in **n**, **o**. The values plotted are the mean ± SD from three independent experiments. The total number of cells analyzed is indicated on the graph (**\*\*\*\***$p < 0.0001$; two-tailed Student's $t$-test). Scale Bars: 10 μm.

modest decrease (~25%) in lysosome-mediated cargo degradation upon RUFY3 depletion, as assessed by BODIPY FL-BSA fluorescence intensity that is de-quenched upon proteolytic cleavage in lysosomes (Fig. 8k–m and Supplementary Fig. 6). These results suggest that although cargo delivery to late endocytic compartments is not affected, RUFY3 depletion likely impacts lysosomal cargo degradation. The impaired degradative ability could be due to reduced lysosome size, as shown in a previous study where lysosomal cargo degradation was less in cells with decreased lysosome size[67]. Furthermore, as Rab7 and Arl8b colocalization is modestly reduced in RUFY3-depleted cells (Fig. 4m), endolysosome formation (generally regarded as the degradative compartments) might be reduced in RUFY3 knockdown.

## Discussion

The small G protein Arl8b is a crucial player regulating lysosomal positioning and functions in the subcellular space[29]. Arl8b overexpression was shown to increase the proportion of lysosomes undergoing bidirectional long-range movement on the microtubule tracks[30]. Subsequent studies revealed that Arl8b binds to effector protein SKIP/PLEKHM2, which in turn binds and recruits kinesin-1 motor to promote anterograde motility of lysosomes[10,31,36,54]. However, it was not known whether Arl8b could mediate the long-range retrograde movement of lysosomes. In this study, we have identified RUFY3 as an Arl8b effector that recruits the JIP4-dynein-dynactin complex to mediate the retrograde motility of lysosomes. Notably, while this work was under review, a preprint study reported similar findings on the role of RUFY3 as an Arl8b effector that promotes dynein-dependent retrograde motility of lysosomes (Tal Keren-Kaplan et al.)[68].

Among the six transcript variants of RUFY3 annotated on NCBI, only variant 2 (469 amino acids long) is functionally characterized and shown to regulate axon guidance in neurons and migration of cancer cells, processes that depend on actin cytoskeletal dynamics[44–48]. This study presents evidence that the longest transcript variant of RUFY3, variant 1 (620 amino acids long), localizes to lysosomes and regulates lysosome positioning. Variant 1 binds to Arl8b via a sequence in its C-terminal region (amino acids 441–561), that is not present in other variants, except for variant 4. Thus, the localization and function of RUFY3 variants may differ based on certain sequence features. As effectors such as PLEKHM1 and SKIP/PLEKHM2 bind to Arl8b via their RUN domains[32], it was surprising that the RUN domain of RUFY3 was not required for binding to Arl8b. Future work is

needed to elucidate what determines the binding of some, but not all, RUN domains to Arl8b.

RUFY3 joins the league of other late endosomal/lysosomal proteins, including RILP, TRPML1, TMEM55B, and SEPT9, which interact with the dynein-dynactin retrograde motor either directly or via binding to dynein adapters JIP3 or JIP4[9,27,41,42]. This list raises a question as to why several dynein adapters are required for lysosomal motility (Fig. 9a). One explanation could be that multiple adapters are needed to engage a sufficient number of dynein motors to win the tug-of-war against kinesin, which generates force equivalent to eight dynein-dynactin complexes[69] (Fig. 9a (i)). A second explanation could be that different adapters are required under different physiological conditions; for instance, one or more lysosomal dynein adapters might be required specifically under conditions such as starvation or oxidative stress where lysosomes are clustered in the perinuclear region (Fig. 9a (ii)). Indeed, the expression of lysosomal adapter TMEM55B is controlled by transcription factors TFEB, TFE3, and SREBF2, activated upon starvation and stress due to cholesterol accumulation in the lysosomal lumen[9]. Additionally, phosphorylation of TMEM55B by ERK/MAPK regulates lysosome positioning[15]. Interestingly, a recent study has shown the involvement of specific dynein adapters at different stages of organelle maturation, providing yet another rationale for the existence of multiple dynein adapters[70].

A third reason could be that while markers like LAMP1 are common, different dynein adapters are essentially required for the motility of distinct compartments (Fig. 9a (iii)). Indeed, recent studies have suggested that there are LAMP1-positive compartments that are non-degradative, and differences in pH and cathepsin activity have been documented between perinuclear and peripheral LAMP1 compartments enriched for Rab7 and Arl8b, respectively[8,71]. Interestingly, a recent study proposed a Rab7-to-Arl8b switch mechanism akin to the Rab5-to-Rab7 switch paradigm for the maturation of late endosomes/endolysosomes[51]. Our data suggest that while RILP is the dynein adapter for Rab7 compartments, RUFY3 is the adapter for compartments enriched for the small G protein Arl8b. The two dynein adapters, RILP and RUFY3, might regulate the positioning of the late endocytic compartments and the maturation/identity of these membranes, and the fate of cargo traffic to and from these compartments. For instance, RILP-mediated Rab7 positioning regulates cargo retrieval from late endosomes, while RUFY3 might promote the close association of Rab7 and Arl8b endosomes and the formation of Rab7-Arl8b hybrid endolysosomal compartments. Eventually, the Rab7-to-Arl8b switch is

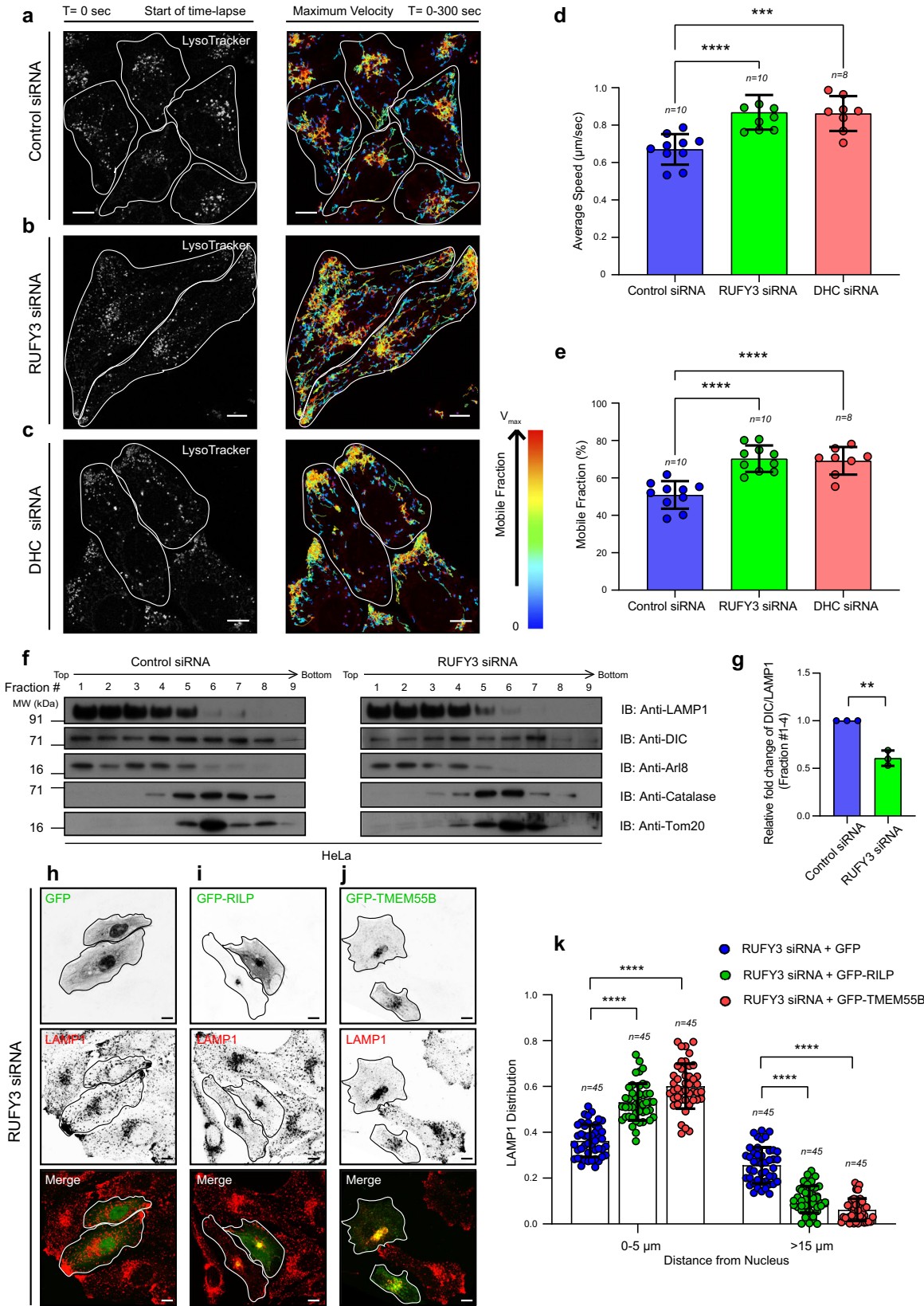

mediated by recruitment of Rab7 GAP TBC1D5 by SKIP, converting a Rab7 and Arl8b hybrid perinuclear compartment to an Arl8b-only peripheral compartment. An exciting question for future studies remains whether Arl8b-binding to RUFY3 regulates its association with the SKIP-Kinesin-1 complex and what physiological cues and molecular players determine the switch

from SKIP-mediated anterograde motility to RUFY3-dependent retrograde motility of lysosomes (Fig. 9b).

While RUFY3 was required for the organization of the lysosome population at the whole-cell scale, surprisingly, its depletion also affected the characteristics of individual lysosomes, namely lysosome size. We found that average lysosome volume was

**Fig. 6 RUFY3 mediates lysosome motility by recruiting dynein motor on lysosomes. a–c** HeLa cells treated with control siRNA (**a**), RUFY3 siRNA (**b**), or DHC siRNA (**c**) were incubated with Lysotracker to label lysosomes. Left panels: representative confocal images of live HeLa cells captured at the start of time-lapse imaging ($T = 0$ sc). Right panels: single-particle tracking analysis of Lysotracker-labeled lysosomes for $T = 300$ s with color-coding to show maximum velocity (blue, immobile; red, max mobility). Scale Bars: 10 μm; see Supplementary Movies 1–3. **d**, **e** The graph represents the maximum average speed (**d**) and a mobile fraction (**e**) of Lysotracker-labeled lysosomes calculated from two independent live-cell imaging experiments as described in **a–c**. The values plotted are the mean ± SD, and the total number of cells analyzed is shown on the graph (****$p < 0.0001$; ***$p < 0.001$; two-tailed Student's $t$-test). **f**, **g** Lysosome enrichment was performed using Optiprep density ultracentrifugation on post-nuclear homogenate prepared from HeLa cells treated with control siRNA or RUFY3 siRNA. Different fractions were resolved and immunoblotted (IB) with the indicated antibodies. The graph depicts the relative fold change in DIC levels normalized to LAMP1 levels (for fractions 1–4) from control and RUFY3 siRNA-treated cells. The values plotted are the mean ± SD from three independent experiments (**$p = 0.0011$; two-tailed Student's $t$-test). **h–j** Representative confocal micrographs of RUFY3 siRNA-treated HeLa cells transfected with GFP (**h**), GFP-RILP (**i**), or GFP-TMEM55B (**j**) and stained for lysosomes using an anti-LAMP1 antibody. Transfected cells are outlined, and some panels are shown in an inverted grayscale. Scale Bars: 10 μm. **k** The distribution of lysosomes based on the LAMP1 signal was quantified from the experiments shown in **h–j**. The values plotted are the mean ± SD from three independent experiments. The total number of cells analyzed is indicated on the graph (****$p < 0.0001$; two-tailed Student's $t$-test).

reduced by a significant value of almost 50% upon RUFY3 depletion. As noted in earlier studies[61,62], a reduction in lysosome size was accompanied by an increase in lysosome number upon RUFY3 depletion. Thus, a significant proportion of lysosomes in RUFY3 knockdown were smaller, numerous, and localized in the peripheral subcellular space. The average velocity of individual lysosomes was increased upon RUFY3 depletion, possibly because the lysosome size was reduced and/or kinesin-mediated forces were dominant on lysosomes (Fig. 9b).

Is there a common explanation that underlies RUFY3 role in regulating the positioning and size of lysosomes? We speculate that in cells depleted of RUFY3, lysosomes escape more frequently from the perinuclear cloud and move in an Arl8b-SKIP-Kinesin-1 complex-dependent manner on the microtubule highway. Additional experimental evidence is required to establish whether kinesin-1-dependent tubulation and fission events, ultimately leading to lysosome reformation, are also enhanced in RUFY3 depletion. An intriguing question is the true identity of the smaller LAMP1-positive vesicles in RUFY3-depleted cells, i.e., whether these are newly formed terminal lysosomes or vesicles retrieving cargo from late endosomes for recycling to the Golgi and plasma membrane? Indeed, previous studies have shown the role of Rab7-retromer and the JIP4-kinesin-1 complex in mediating tubulation and cargo retrieval from late endosomes[72–74]. From this study, we speculate that the RUFY3 role is more likely to be downstream of the late endosomal sorting step and in maintaining the balance between terminal storage lysosomes and endolysosomes.

Future studies will establish whether the correlation between lysosome positioning and size reflects different biogenesis stages of this enigmatic organelle with newly formed immature lysosomes located in the cell periphery. In contrast, mature lysosomes reside in the perinuclear pool, poised for fusion with incoming cargo vesicles.

## Methods

**Cell culture and treatments**. HeLa, HEK293T, U2OS, and A549 cells (from ATCC) were maintained in DMEM media (Gibco) supplemented with 10% FBS (Gibco) at 37 °C with 5% $CO_2$ in a humidified cell culture chamber. For imaging and flow cytometry experiments described below, phenol red-free DMEM media (Gibco) was used. For culturing ARPE-19 cells (from ATCC), DMEM/F-12 media (Gibco) supplemented with 10% FBS was used. Serum starvation was performed by incubating cells in DMEM with 2 mM L-glutamine for 1 h. Combining amino acid and serum starvation was performed by incubating cells in EBSS for 4 h. Amino acid starvation was performed by incubating cells in amino acid-free DMEM (US Biologicals) supplemented with 10% dialyzed-FBS (Gibco) for 4 h. Each cell type was regularly screened for the absence of mycoplasma contamination by using the MycoAlert Mycoplasma Detection Kit (Lonza) and was cultured for no more than 15 passages.

For gene silencing, siRNA oligos or SMARTpool were purchased from Dharmacon and prepared according to the manufacturer's instructions. Following siRNA oligos

were used in this study: control siRNA, 5′-TGGTTTACATGTCGACTAA-3′; RUFY3 siRNA, 5′-GATGCCTGTTCAACAAATGAA-3′; Arl8b siRNA, 5′-AGGT AACGTCACAATAAAGAT-3′; Rab7a siRNA, 5′-CTAGATAGCTGGAGAGATG-3′; JIP4 siRNA, 5′-GAGCATGTCTTTACAGATC-3′; DHC siRNA, 5′-GAGAGGAGG TTATGTTTAA-3′; PIKFYVE siRNA, ON-TARGETplusSMARTpool (L-005058-00-0005). For shRNA-mediated gene silencing, control shRNA (SHC016) and RUFY3 shRNA (TRCN0000127915) were purchased from Sigma-Aldrich. Transient transfection of siRNAs was performed with DharmaFECT 1 (Dharmacon) according to the manufacturer's instructions.

For shRNA-mediated gene silencing, lentiviral transduction was performed as described previously[52]. Briefly, for lentiviral transduction, HeLa cells were plated at 100,000/well in six-well plates (Corning) in 8 μg/mL Polybrene (Sigma-Aldrich) and transduced by addition of 100 μL viral supernatant. 24 h later, puromycin (Sigma-Aldrich) was added at 3 μg/mL to select transductants and experiments performed on days 5-21 following transduction.

**Mammalian expression constructs**. All the expression plasmids used in this study are listed in Supplementary Table 1.

**Antibodies and chemicals**. All the antibodies used in this study are listed in Supplementary Table 2. Alexa-Fluor-conjugated-Dextran, Lysotracker dyes, Lysosensor dyes, BODIPY FL LDL, Phalloidin, and DAPI, were purchased from Molecular Probes (Invitrogen). SiR-Lysosome Kit was purchased from Cytoskeleton, Inc. Self-Quenched BODIPY FL conjugate of BSA was purchased from BioVision. Polybrene, Puromycin, EBSS, Rapamycin, and Bafilomycin A1 were purchased from Sigma-Aldrich.

**Transfection, immunofluorescence, and live-cell imaging**. Cells grown on glass coverslips (VWR) were transfected with desired constructs using X-treme GENE-HP DNA transfection reagent (Roche) for 16–18 h. Cells were fixed in 4% PFA in PHEM buffer (60 mM PIPES, 10 mM EGTA, 25 mM HEPES, 2 mM $MgCl_2$, and final pH 6.8) for 10 min at room temperature (RT). Post-fixation, cells were incubated with blocking solution (0.2% saponin + 5% normal goat serum (NGS) in PHEM buffer) at RT for 30 min, followed by three washes with 1X PBS. Following the blocking step, cells were incubated with primary antibodies in staining solution (PHEM buffer + 0.2% saponin + 1% NGS) for 1 h at RT, washed three times with 1X PBS, and then incubated for 30 min at RT with Alexa-fluorophore-conjugated secondary antibodies in staining solution. Coverslips were mounted using Fluoromount G (Southern Biotech), and confocal images were acquired using Carl Zeiss 710 Confocal Laser Scanning Microscope with a Plan Apochromat 63×/1.4 NA oil immersion objective and high-resolution microscopy monochrome cooled camera AxioCamMRm Rev. 3 FireWire (D) (1.4 megapixels, pixel size 6.45 μm × 6.45 μm). ZEN 2012 v. 8.0.1.273 (ZEISS) software was used for image acquisition. All images were captured to ensure that little or no pixel saturation was observed. The representative confocal images presented in figures were processed and adjusted for brightness and contrast using Fiji software[75] or Adobe Photoshop CS.

To minimize the fluorescent signal from the cytosolic pool of overexpressed protein, the cells were permeabilized for 5 min on ice with 0.05% saponin in PHEM buffer before the fixation step as described in refs. [76–78]. This method was performed for experiments shown in Figs. 1i, l–n; 4f, g; and Supplementary Fig. 2c–g, j, k.

To label lysosomes with Lysotracker or SiR-Lysosome probes, uptake was done as per the manufacturer's instructions. Briefly, cells were incubated in phenol red-free complete DMEM media (Gibco) containing SiR-Lysosome (1 μM) or Lysotracker Deep Red (100 nM) for 1 h at 37 °C in a cell culture incubator. Cells were washed three times with 1X PBS to remove excess probe followed by fixation with 4% PFA in PHEM buffer as described above.

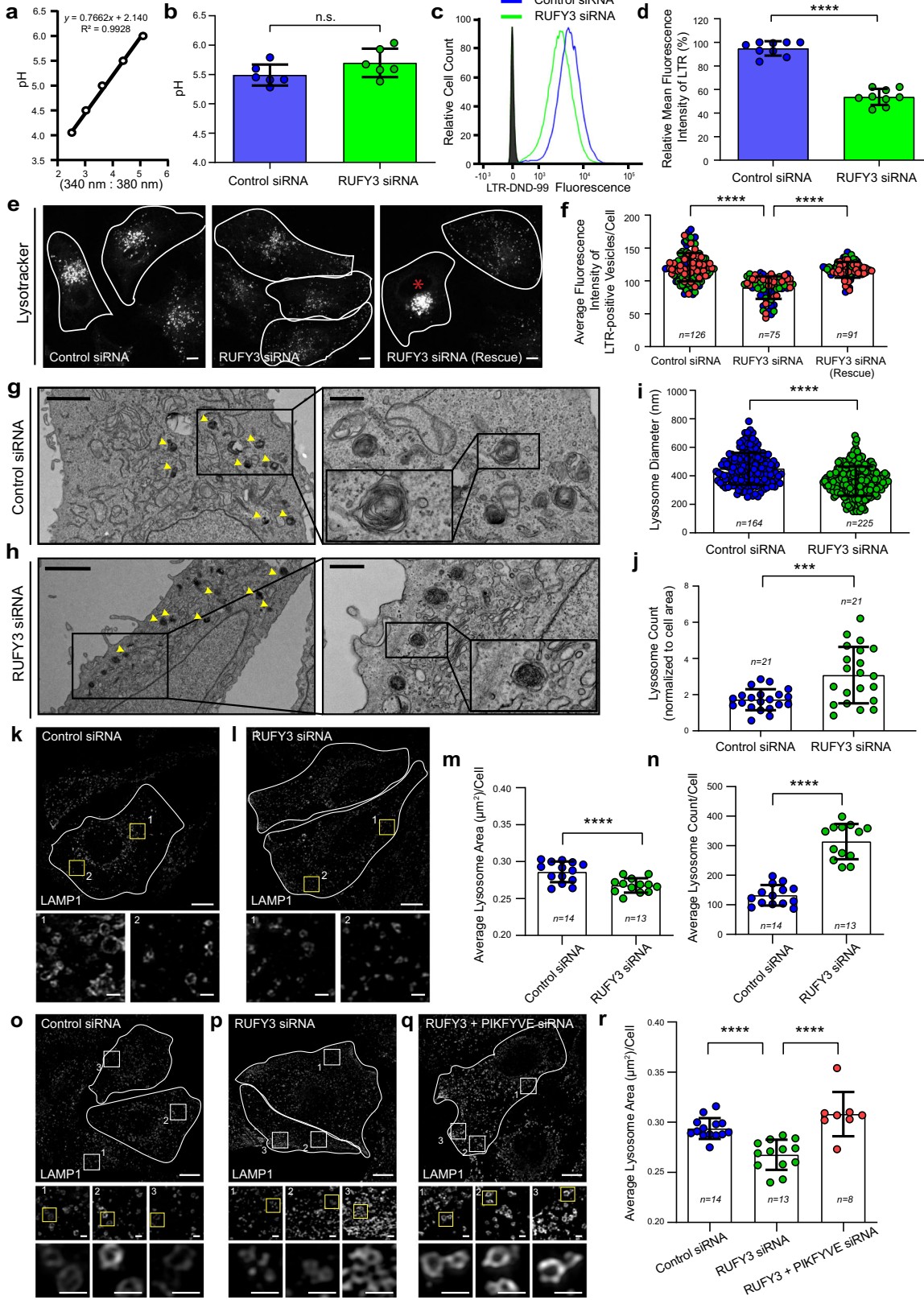

For live-cell imaging experiments, cells were seeded on glass-bottom tissue culture treated cell imaging dish (Eppendorf). For vesicle tracking experiments, cells were incubated in phenol red-free complete DMEM media containing Lysotracker (LTR DND-99; 100 nM) for 10 min at 37 °C in a cell culture incubator. Live-cell imaging was performed using Zeiss LSM 710 confocal microscope equipped with an environmental chamber set at 37 °C and 5% $CO_2$.

**Structured illumination microscopy (SIM)**. For SIM imaging, cells were processed, fixed, and stained as described previously. SIM images were captured with Zeiss ELYRA 7 (Lattice SIM Technology) using either Plan Apo 40×/1.40 oil or Plan Apo 63×/1.40 oil objective and sCMOS camera (PCO Edge). A lattice pattern structured samples and 15 phases shifted raw images were acquired for every Z plane with a slice size of 110 nm. The complete system control, imaging and

**Fig. 7 RUFY3 depletion reduces lysosome size. a** pH calibration curve based on ratiometric fluorescence intensity measurements of Lysosensor Yellow/Blue DND-160. **b** Graph showing average pH value of lysosomes measured from HeLa cells treated with the indicated siRNAs. Values plotted are the mean ± SD from six independent experiments (n.s. not significant; two-tailed Student's t-test). **c, d** Representative histogram showing mean fluorescence intensity (MFI) of Lysotracker Red DND-99 (LTR) uptake (1 h) in control siRNA- and RUFY3 siRNA-treated HeLa cells (**c**), and the graph in **d** represents the relative percentage of MFI for LTR uptake from three independent experiments (****$p < 0.0001$; two-tailed Student's t-test). **e, f** Representative micrographs of live HeLa cells treated with the indicated siRNAs and labeled with LTR. The asterisk indicates cells transfected with the GFP-RUFY3 siRNA-resistant plasmid. Scale Bars: 10 μm. The quantification of the average fluorescence intensity of LTR-positive vesicles is shown in **f**. The values plotted are the mean ± SD from three independent experiments. Experiments are color-coded, and the total number of cells analyzed is indicated on the graph (****$p < 0.0001$; two-tailed Student's t-test). **g–j** Representative TEM images of HeLa cells treated with the indicated siRNAs. Higher magnifications of lysosomes (dense and multi-lamellar structures, indicated by yellow arrowheads) are shown in the right panels. Scale Bars: 2 μm (main); 0.5 μm (inset). Lysosome size (**i**) and number (**j**) were quantified using TEM images. Note: in **i**, $n$ represents the number of lysosomes analyzed for size measurement. **k–n** Representative SIM images of HeLa cells treated with indicated siRNAs and stained with anti-LAMP1 antibodies. Insets represent a magnified view of boxed areas, highlighting differences in lysosome size. The average area (**m**) and count (**n**) of LAMP1-positive vesicles per cell was measured in HeLa cells upon treatment with the indicated siRNAs. **o–r** Representative SIM images of HeLa cells treated with the indicated siRNAs and stained with anti-LAMP1 antibody. In the insets, zoomed views of selected ROIs are shown, and quantification of the average area of lysosomes is plotted (**r**). The values plotted are the mean ± SD and the total number of cells analyzed are indicated on the graph (****$p < 0.0001$; ***$p = 0.0006$; two-tailed Student's t-test). Scale Bars: 10 μm (main); 1 μm (inset).

processing of raw image files to final super-resolution images were done using the SIM module of the Zen Black v. 3.0 SR (Zeiss) software (Carl Zeiss MicroImaging).

**Image analysis and quantification**

*Analysis of lysosome distribution.* To quantify the distribution of lysosomes based on LAMP1/Lysotracker/SiR-Lysosome signal intensity, Fiji software was used. A boundary was drawn along the periphery of each selected cell using the freehand selection tool. With the clear outside function of Fiji software, removed LAMP1 signals from nearby cells. Next, an ROI was drawn around the nucleus (using DAPI fluorescence signal), and LAMP1 signal intensity was measured for that section. The same ROI was then incremented by 5 μm till the cell periphery, and LAMP1 intensity was measured for each incremented ROI. Finally, LAMP1 intensity was calculated for perinuclear (0–5 μm; by subtracting the intensity of the first ROI from second) and periphery (>15 μm; by subtracting the intensity of the fourth ROI from total cell intensity) region of cell as shown in Fig. 2d. LAMP1 distribution was plotted by dividing each section's intensity (perinuclear and periphery) with whole-cell LAMP1 intensity. The same methodology was employed for quantifying mitochondria distribution (based on TOM-20 signal intensity) from images presented in Fig. 3i.

The analysis of lysosome distribution was also performed by measuring the fractional distance of lysosomes from the cell center using the plot profile tool of Fiji software as described previously[5,33]. Briefly, in a confocal micrograph, a line was drawn from the center of the nucleus to the periphery of the cell. Next, using the plot profile tool, all the lysosomal marker fluorescent intensities and their corresponding distance values along the line were extracted. After determining the signal threshold, background pixels and their corresponding distances were excluded from the analysis. All the remaining distances (corresponding to the lysosomes pixels only) were converted to fractional distance by dividing all the values by the total distance of the line as shown in Supplementary Fig. 2l. The same methodology was employed for quantifying mitochondria distribution (based on TOM-20 signal intensity) from images presented in Supplementary Fig. 3k.

*Analysis of LAMP1 and Lysotracker-positive vesicles.* To measure the area and number of LAMP1-positive vesicles from SIM images, Z stacks of each micrograph was converted to 8-bit Max Intensity Projection using Fiji software. Using the Analyze Particle tool with the Otsu threshold was used for calculating the area and number. For TEM micrographs, the diameter of individual lysosomes was measured manually by drawing a straight line across the lysosome using the Line tool in Fiji software. For analyzing Lysotracker intensity from confocal micrographs, Fiji software was used.

*Surface area analysis.* The surface area of cells was quantified manually by drawing the periphery of the cell (using Phalloidin staining) using the Freehand and Measure Function tools in Fiji software.

*Colocalization analysis.* For all the colocalization analysis, the JACoP plugin of Fiji software was used to determine Pearson's correlation coefficient and Mander's overlap.

**Single-particle tracking.** To perform a single-particle tracking analysis of lysosomes, cells were incubated with Lysotracker 100 nM (LTR DND-99) for 10 min at 37 °C in phenol red-free complete DMEM media. Time-lapse confocal imaging was done as discussed above. To measure mobile fraction and the average speed of

lysosomes from time-lapsed images, the TrackMate plugin[79] of Fiji software was used with the following parameters:

Vesicle diameter, 1 μm
Detector, DoG
Initial thresholding, none
Tracker, Simple LAP tracker
Linking max distance, 2 μm
Gap-closing max distance, 2 μm
Gap-closing max frame gap, 2
Filters, none

Data were exported to a Microsoft Excel spreadsheet (2013) for further analysis.

**Cell lysates, co-immunoprecipitation, and immunoblotting.** For preparing lysates, cells were lysed in ice-cold RIPA lysis buffer (10 mM Tris-Cl (pH 8.0), 1 mM EDTA, 0.5 mM EGTA, 1% Triton X-100, 0.1% SDS, 0.1% sodium deoxycholate, 140 mM NaCl supplemented with phosstop (Roche), and protease inhibitor cocktail (Sigma-Aldrich)). The samples were incubated on ice for 2 min followed by vortexing for 30 s, and this cycle was repeated a minimum of five times and subjected to centrifugation at 16,627×$g$ for 10 min at 4 °C. The clear supernatants were collected, and protein amounts were quantified using the BCA kit (Sigma-Aldrich).

To perform co-immunoprecipitation, cells were lysed in ice-cold TAP lysis buffer (20 mM Tris pH 8.0, 150 mM NaCl, 0.5% NP-40, 1 mM MgCl$_2$, 1 mM Na$_3$VO$_4$, 1 mM NaF, 1 mM PMSF, and protease inhibitor cocktail). The lysates were incubated with indicated antibody-conjugated-agarose beads at 4 °C rotation for 3 h, followed by four washes with TAP wash buffer (20 mM Tris pH 8.0, 150 mM NaCl, 0.1% NP-40, 1 mM MgCl$_2$, 1 mM Na$_3$VO$_4$, 1 mM NaF, and 1 mM PMSF). The samples were loaded on SDS-PAGE for further analysis.

For immunoblotting, protein samples separated on SDS-PAGE were transferred onto PVDF membranes (Bio-Rad). Membranes were blocked overnight at 4 °C in blocking solution (10% skim milk in 0.05% PBS-Tween 20). Indicated primary and secondary antibodies were prepared in 0.05% PBS-Tween 20. The membranes were washed for 10 min thrice with 0.05% PBS-Tween 20 or 0.3% PBS-Tween 20 after 2 h incubation with primary antibody and 1 h incubation with secondary antibody. The blots were developed using a chemiluminescence-based method (Thermo Scientific) using X-ray films (Carestream). To perform densitometry analysis of immunoblots, Fiji software was used as described in ref. [80].

**Recombinant protein purification, GST-pulldown assay, and mass spectrometry analysis.** All the recombinant proteins used in this study were expressed and purified in the *E. coli* BL21 strain (Invitrogen). A single transformed colony was inoculated in Luria–Bertani broth containing plasmid vector antibiotic and incubated at 37 °C in a shaking incubator for setting-up primary cultures. Following 8–12 h of culturing, 1% of primary inoculum was used to set up secondary cultures and subjected to incubation at 37 °C with shaking until absorbance of 0.6 at 600 nm was reached. For induction of protein expression, 0.3 mM IPTG (Sigma-Aldrich) was added to the cultures, followed by incubation for 16 h at 16 °C with shaking. Post-induction period, bacterial cultures were centrifuged at 3542×$g$ for 10 min, washed once with 1XPBS, and resuspended in lysis buffer (20 mM Tris and 150 mM NaCl, pH 7.4) containing protease inhibitor tablet (Roche) and 1 mM PMSF (Sigma-Aldrich). Bacterial cells were lysed by sonication, followed by centrifugation at 15,557×$g$ for 30 min at 4 °C. The clear supernatants were incubated with glutathione resin (Gbiosciences) to allow binding of GST-tagged proteins or His60 Ni Superflow resin (Takara) for binding of His-tagged proteins on rotation

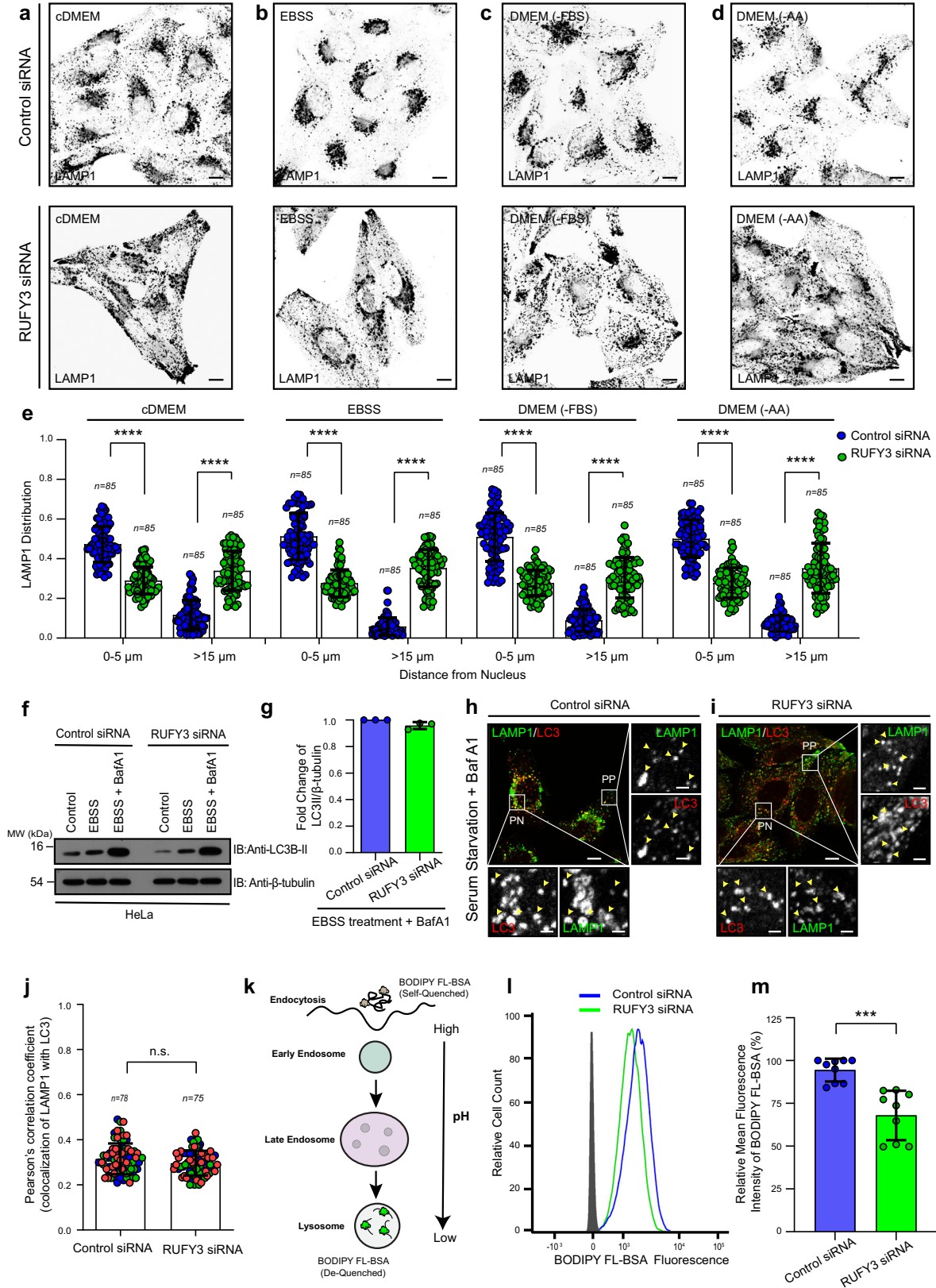

for 1–2 h at 4 °C The beads were washed a minimum of six times with wash buffer (20 mM Tris, 300 mM NaCl, pH 7.4) to remove impurities.

For semi-purified preparation of FLAG-tagged-JIP4 from mammalian cells, HEK293T cells transfected with FLAG-JIP4 expressing construct were lysed in NP-40 buffer (30 mM HEPES pH 7.4, 50 mM Potassium acetate, 2 mM Magnesium acetate, 1 mM EGTA, 10% Glycerol, 5 mM DTT, 0.1% NP-40, 1 mM PMSF, protease inhibitor cocktail) by performing three rounds of the freeze-thaw cycle. To carry out this step, cells were incubated on dry ice for 10 min and then transferred to ice-cold water for 10 min. The cell lysate was centrifuged at 20,000×g for 20 min at 4 °C, and the cleared lysate was incubated with anti-FLAG antibody-conjugated-agarose beads (Biolegend) for 3 h at 4 °C on rotation. Beads were washed three times with lysis buffer by incubating for 5 min at 4 °C on rotation. FLAG-JIP4 was eluted from the beads using a FLAG-peptide (Sigma-Aldrich) at a final concentration of 340 µM in lysis buffer.

**Fig. 8 RUFY3 regulates nutrient-dependent lysosome repositioning. a–e** Representative confocal micrographs (shown as grayscale inverted) of HeLa cells treated with control siRNA or RUFY3 siRNA and incubated in the indicated media for 4 h. Post-treatment, cells were fixed and stained using an anti-LAMP1 antibody. The distribution of lysosomes based on the LAMP1 signal from these experiments is shown in **e**, and the values plotted are the mean ± SD from three independent experiments, and the total number of cells analyzed is indicated on the graph (****$p < 0.0001$; two-tailed Student's $t$-test). **f, g** HeLa cells transfected with indicated siRNAs were grown in complete media or subjected to 2 h starvation using EBSS media in the absence or presence of Bafilomycin A1 (BafA1). Lysates from these cell types were IB with the indicated antibodies. Protein densitometric analysis of LC3B-II levels normalized to β-tubulin is shown in **g**. **h, i** Representative confocal images of control (**h**) and RUFY3-depleted (**i**) HeLa cells incubated in media lacking serum for 1 h in the presence of BafA1. Post-treatment, cells were fixed and stained for LAMP1 and LC3. In the insets, selected peripheral (PP) and perinuclear (PN) regions of the cell are magnified to show colocalized pixels of LC3 with LAMP1 (denoted by yellow arrowheads). **j** Colocalization of LAMP1 with LC3 for the experiments performed in **h** and **i** was analyzed by measuring Pearson's correlation coefficient. The values plotted are the mean ± SD from three independent experiments. Experiments are color-coded, and the total number of cells analyzed is on the graph (n.s. not significant; two-tailed Student's $t$-test). **k** Schematic representation of the BODIPY FL-BSA uptake and de-quenching in lysosomes. **l, m** Representative histogram showing mean fluorescence intensity (MFI) of de-quenched BODIPY FL-BSA after 2 h of incubation in control siRNA- and RUFY3 siRNA-treated HeLa cells as analyzed by flow cytometry (**l**), and the bar graph in **m** represents the relative percentage of MFI signal for de-quenched BODIPY FL-BSA after 2 h of incubation in HeLa cells treated with control- or RUFY3 siRNA calculated from three independent experiments (***$p < 0.001$; two-tailed Student's $t$-test). Scale Bars: 10 μm (main); 2 μm (inset).

For GST-pulldown assay using mammalian cells as a source of lysates, cells were lysed in ice-cold TAP lysis buffer (20 mM Tris (pH 8.0), 150 mM NaCl, 0.5% NP-40, 1 mM MgCl$_2$, 1 mM Na$_3$VO$_4$, 1 mM NaF, 1 mM PMSF, and protease inhibitor cocktail), followed by incubation in ice for 10 min and centrifuged at 16,627×$g$ for 10 min. Lysates were collected and incubated with GST or GST-tagged proteins bound to glutathione resin at 4 °C for 3–4 h with rotation. Following incubation, beads were washed a minimum of six times with TAP lysis buffer, and elution was done by boiling the samples in Laemmli buffer and subjected to SDS-PAGE for further analysis.

For GST-pulldown experiments using purified proteins, recombinant His-Arl8b, GST, and GST-tagged proteins were quantified using a BCA protein assay kit (Sigma-Aldrich). Five micrograms of GST (as a control) and GST-tagged proteins were bound to glutathione beads for 3 h at 4 °C on rotation. The beads were blocked with 5% BSA for 2 h at 4 °C on rotation to prevent nonspecific binding. The beads were washed with TAP lysis buffer (20 mM Tris (pH 8.0), 150 mM NaCl, 0.5% NP-40, 1 mM MgCl$_2$, 1 mM Na$_3$VO$_4$, 1 mM NaF, 1 mM PMSF, and protease inhibitor cocktail) minimum three times and incubated with 5 μg of His-Arl8b at 4 °C for 1 h with rotation. After binding, the beads were washed five times with TAP lysis buffer followed by elution in 4x Laemmli buffer and SDS-PAGE for further analysis. A similar protocol was followed for performing binding assay between semi-purified preparations of FLAG-JIP4 with GST and GST-RUFY3 proteins except for the use of NP-40 buffer in place of TAP lysis buffer.

To search for potential interacting partners of RUFY3, GST-pulldown assay followed by identification of proteins using mass spectrometry was done as described[32,52]. Briefly, recombinant GST-RUFY3 and GST-only (as a control) proteins were used as bait proteins and incubated with lysates prepared from HEK293T cells as described above. The coomassie stained protein bands that were specifically present in the GST-RUFY3 sample lane (Sample A: ~250–280 kDa band; Sample B: ~200–250 kDa; Sample C: ~80 kDa; and Sample D: ~16 kDa band) were cut out and submitted to Taplin Mass Spectrometry Facility (Harvard Medical School, Boston, USA) for protein identification. As a control, the whole GST-only sample lane was cut out to identify proteins that might be binding to GST protein only. The complete list of prey proteins (and their peptide counts) identified in each sample set is provided in Supplementary Data 1. RAW data are available via ProteomeXchange consortium with identifier PXD027010.

**Yeast two-hybrid assay**. Matchmaker Gold Yeast Two-Hybrid System (Clontech) was used as per the manufacturer's instructions for carrying out yeast two-hybrid screening. Briefly, human Arl8b cDNA cloned in GAL4-BD vector (pGBKT7) was used as bait. The bait plasmid transformed Y2HGold yeast strain was mated with Y187 strain transformed with human brain cDNA library. A small-scale yeast two-hybrid assay was carried out as described previously[81]. Briefly, plasmids encoding GAL4-AD and GAL4-BD fusion encoding constructs were co-transformed in *Saccharomyces cerevisiae* Y2HGold strain (Clontech), streaked on plates lacking leucine and tryptophan (-Leu/-Trp), and allowed to grow at 30 °C for 3 days. The cotransformants were replated on a nonselective medium (-Leu/-Trp) and selective medium (-Leu/-Trp/-His) to assess interaction. All the drop-out yeast media was purchased from Takara.

**Lysosome immunoisolation**. To immunopurified lysosomes, the Lyso-IP method was used with some modifications[50]. HEK293T cells stably expressing TMEM192-FLAG (control) or TMEM192-HA were collected and resuspended in ice-cold KPBS (136 mM KCl, 10 mM KH$_2$PO$_4$, adjusted to pH 7.25 with KOH) buffer and homogenized using dounce homogenizer (~20 strokes). The homogenized cells were gently collected and centrifuged for 2 min at 1000×$g$. The supernatant obtained was incubated with anti-HA antibodies-conjugated-agarose beads (Sigma-Aldrich) at 4 °C for 15 min. Beads were gently washed thrice with KPBS, and bound lysosomes were eluted in Laemmli buffer and subjected to SDS-PAGE for further analysis.

**Subcellular fractionation**. To perform lysosome enrichment, subcellular fractionation was carried out using the Lysosome Enrichment Kit (Thermo Scientific). Briefly, the cell pellet was resuspended in PBS and homogenized with a dounce homogenizer on ice (~20 strokes). To confirm cell lysis, microscopic examination of homogenate was done by adding 0.5% trypan dye. The homogenate was subjected to centrifugation at 500×$g$ for 10 min at 4 °C, and post-nuclear supernatant (PNS) was diluted in OptiPrep gradient media (Sigma-Aldrich) to a final concentration of 15% OptiPrep. The sample was then carefully overlayered on the top of a discontinuous density gradient (17, 20, 23, 27, and 30%). The gradient was subjected to ultracentrifugation at 145,000×$g$ in an SW60 Ti swinging bucket rotor (Beckman Coulter) for 4 h at 4 °C. After the spin, eight fractions of 400 μl each were collected from top to bottom. The fractions were spun again at 18,000×$g$ for 20 min in an SW41 Ti rotor at 4 °C, and the resulting pellet was suspended in 4X SDS-sample buffer, boiled for 10 min, and analyzed by SDS-PAGE and immunoblotting.

**Measurement of lysosome pH**. To measure the lysosome's pH, Lysosensor Yellow/Blue DND-160 was used as described previously[58]. Briefly, cells were trypsinized and incubated with 2 μM Lysosensor Yellow/Blue DND-160 (Invitrogen) for 3 min at 37 °C in phenol red-free complete DMEM media. Cells were rinsed twice with 1X PBS to remove excess dye and incubated for 10 min in isotonic pH calibration buffers (143 mM KCl, 5 mM Glucose, 1 mM MgCl$_2$, 1 mM CaCl$_2$, 20 mM MES, 10 μM Nigericin, and 5 μM Monensin) ranging from 4 to 6. Next, ~10,000 cells/well were distributed into a black 96-well plate (Thermo Scientific), and fluorescence reading was recorded at 37 °C using a 96-well plate multi-mode fluorescence reader (Tecan Infinite M-PLEX). Samples were excited at 340 and 380 nm wavelengths to detect emitted light at 440 and 540 nm, respectively. The pH calibration curve was generated by plotting the fluorescence intensity ratio of 340 to 380 nm against the respective pH value of buffers.

**Flow cytometry**. To quantify Lysotracker uptake, cells were incubated in phenol red-free complete DMEM media (Gibco) containing 100 nM Lysotracker Red (LTR DND-99; Invitrogen) for 1 h at 37 °C. Post-incubation period, media was removed, and cells were trypsinized, washed, and resuspended in ice-cold 1X PBS and analyzed by flow cytometry. To measure the proteolytic activity of lysosomes, cells were incubated in phenol red-free complete DMEM media (Gibco) containing 20 μg/mL BODIPY FL-BSA (BioVision) for 2 h at 37 °C. Post-incubation period, media was removed, and cells were trypsinized, washed, and resuspended in ice-cold 1X PBS and analyzed by flow cytometry. Sample acquisition was done with BD FACS Aria Fusion Cytometer using BD FACS Diva software version 8.0.1 (BD Biosciences). Data analysis was done using BD FlowJo version 10.0.1.

**Dextran trafficking assay**. Dextran delivery to lysosomes was performed as described with some minor modifications[52]. Briefly, to label lysosomes, control and RUFY-silenced HeLa cells were incubated with Lysotracker Deep Red (100 nM) containing phenol red-free complete DMEM media for 10 min at 37 °C. Cells were further incubated with dextran (Alexa-Fluor 488-conjugated-dextran; green) for 1 and 2 h at 37 °C. At the end of the incubation period, cells were washed with 1X PBS followed by fixation and mounting as described earlier. The coverslips were imaged immediately by confocal microscopy. The colocalization of dextran with Lysotracker-labeled lysosomes was assessed using the JACoP plugin of Fiji software.

**LDL trafficking assay**. For LDL trafficking assay, control and RUFY3-silenced HeLa cells seeded on live-cell imaging dishes were starved for 8 h in DMEM media containing 5% charcoal-stripped FBS (Gibco) (starvation media). The cells were

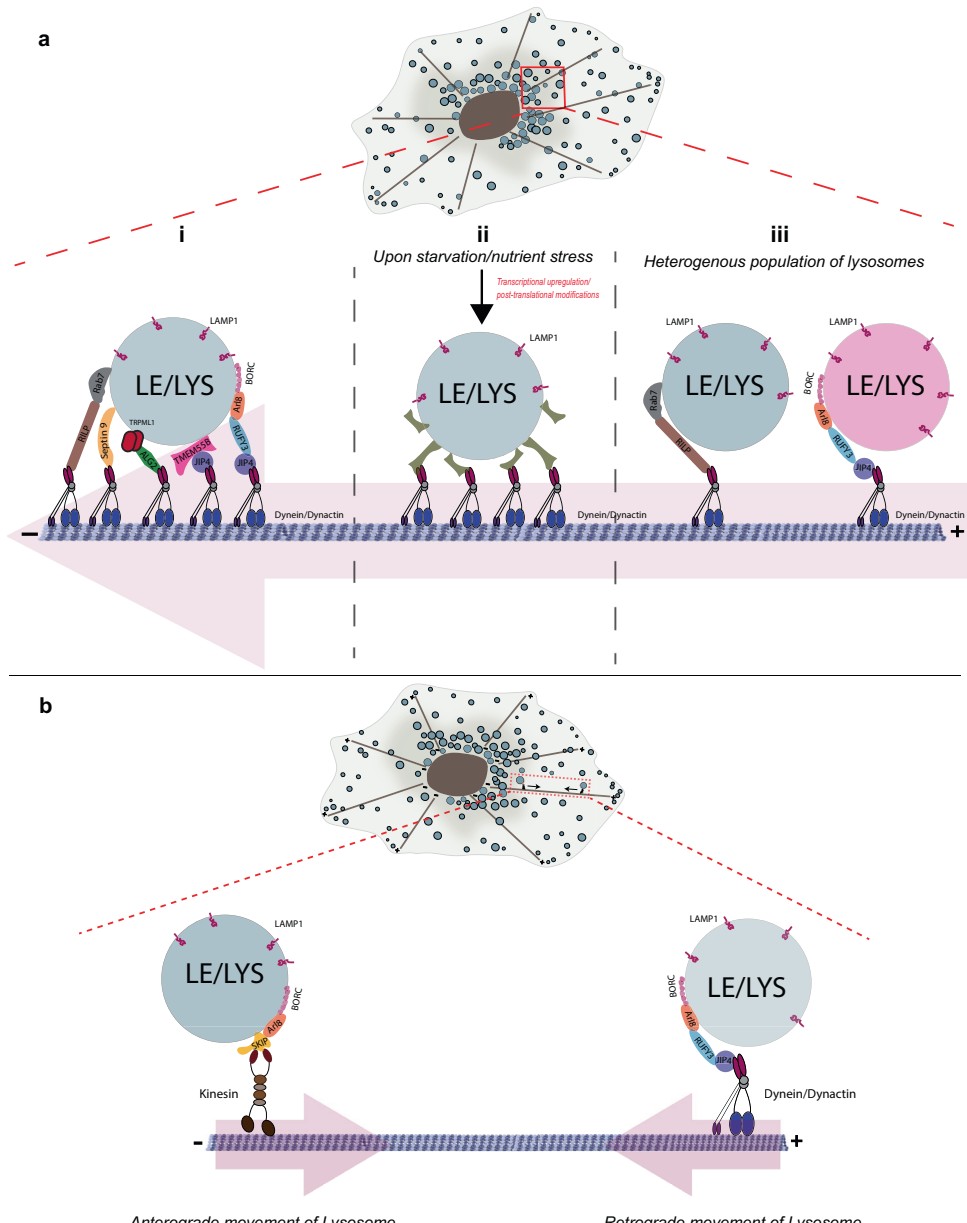

**Fig. 9 Schematic representing the role of multiple motor-adaptors in lysosome positioning. a** Three distinct hypothetical scenarios to explain the significance of different lysosomal adapters that engage the dynein-dynactin complex for retrograde transport. (**i**) Multiple adapters may work in concert to recruit enough dynein motors to balance the opposite driving forces exerted by a single kinesin motor. (**ii**) Different adapters may be required under distinct physiological conditions. For instance, during nutrient starvation, expression, and/or recruitment of a particular adapter might increase onto lysosomes. Increased spatial density of lysosomes and autophagosomes in the perinuclear region could enhance their fusion. (**iii**) Different dynein adapters are required for retrograde transport of distinct populations of lysosomes that may differ in their membrane composition. **b** A model illustrating the opposing motor adapters recruited by Arl8b. Arl8b relieves autoinhibition of SKIP, which in turn recruits and activates the Kinesin-1 motor to promote anterograde motility of lysosomes. As revealed in this study, Arl8b recruits RUFY3 on lysosomes, which then interacts with the JIP4-dynein-dynactin complex to mediate lysosome retrograde movement.

then pulsed with BODIPY FL LDL (7.5 µg/mL; Invitrogen) made in starvation media for 10 min. The cells were washed with 1X PBS and chased in phenol red-free complete media containing Lysotracker Red DND-99 (100 nM) to label lysosomes. Time-lapse confocal imaging was done at 0, 30, 60, and 120 min of the chase. The colocalization between LDL and Lysotracker at different time periods was measured using the JACoP plugin of Fiji software.

**Autophagy flux assay**. Autophagic flux was determined by checking for the rescue of LC3B-II degradation by treating HeLa cells with V-ATPase inhibitor Bafilomycin A1 (100 nM; Sigma-Aldrich) at steady-state or with serum starvation in

EBSS for 2 h. After treatment, cells were lysed using ice-cold RIPA buffer supplemented with protease inhibitor. An equal amount of lysates were loaded on SDS-PAGE, transferred to PVDF membrane, and probed for LC3B-II and β-tubulin. Densitometry analysis of LC3B-II band intensity normalized to β-tubulin intensity was done using Fiji software.

**Transmission electron microscopy (TEM)**. Sample processing and TEM were performed at the Harvard Medical School EM Facility (Boston, USA). Briefly, HeLa cells transfected with control siRNA or RUFY3 siRNA were fixed in routine fixative (2.5% glutaraldehyde/1.25% paraformaldehyde in 0.1 M sodium cacodylate buffer,

pH 7.4) for 1 h at RT and washed in 0.1 M sodium cacodylate buffer (pH 7.4). The cells were then postfixed for 30 min in 1% osmium tetroxide/1.5% potassium ferrocyanide, washed with water three times, and incubated in 1% aqueous uranyl acetate for 30 min, followed by two washes in water and subsequent dehydration in grades of alcohol (5 min each: 50, 70, 95, 2 × 100%). Cells were removed from the dish in propylene oxide, pelleted at 1741×$g$ for 3 min, and infiltrated overnight in a 1:1 mixture of propylene oxide and TAAB Epon (Marivac Canada). The samples were subsequently embedded in TAAB Epon and polymerized at 60 °C for 48 h. The ultrathin sections were cut on a Reichert Ultracut-S microtome, picked up onto copper grids stained with lead citrate, and examined in a JEOL 1200EX transmission electron microscope having an AMT 2k charge-coupled device camera.

**Statistics and reproducibility.** Graphs represent mean ± SD and $p$ values were calculated using two-tailed Student's $t$-test (GraphPad Prism 8.0). Differences between groups were considered statistically significant for $p$ values < 0.05. All experimental data shown in this report, including immunofluorescence micrographs, were analyzed from at least three independent experiments or at least eight cells.

**Reporting summary.** Further information on research design is available in the Nature Research Reporting Summary linked to this article.

## Data availability

Raw data files of mass spectrometry results were deposited onto ProteomeXchange consortium with identifier PXD027010. All relevant data supporting this study's findings are presented in the manuscript and supplementary information. Raw data, uncropped Western blots, and yeast two-hybrid plates scan are available in the Source Data file which is provided with the manuscript. Source data are provided with this paper.

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

## Acknowledgements

G.K. and N.D. acknowledge fellowship support from CSIR. P.C. acknowledges fellowship support from IISER Mohali. This work was supported by the Department of Biotechnology (DBT)/Wellcome Trust India Alliance Intermediate Fellowship (IA/I/14/2/501543) to A.T. and Senior Fellowship (IA/S/19/1/504270) to M.S. The authors acknowledge Dr. Nitin Mohan (IIT Kanpur) for advice on single-particle tracking-related experiments and Drs. Marlieke Jongsma, Ilana Berlin, and Carlos Guardia for their helpful suggestions in performing analysis of lysosome distribution. The authors would also like to acknowledge Prateek Arora (IISER Mohali FACS Facility) for technical help in flow cytometry, Maria Ericson (Harvard Medical School) for EM imaging, Ross Tomaino (Taplin MS Facility, Harvard Medical School) for mass spectrometry analysis, and all the lab members of M.S. and A.T. labs for their advice and support. CSIR-IMTECH Communication No. 016/2021.

## Author contributions

G.K., M.S., and A.T. conceived and designed the project. G.K. performed the majority of the experiments, analyzed the results, and prepared the figures. P.C. performed the FRB-FKBP based experiments and some lysosome positioning-related experiments. N.D. helped in performing live-cell imaging and flow cytometry experiments during the manuscript revision stage. S.C., S.S., and K.S. conducted the protein–protein interaction experiments and provided critical molecular biology reagents. G.K., M.S., and A.T. wrote the manuscript.

## Competing interests

The authors declare no competing interests.
