## [Peer Review File · Nature Communications]

RUFY3 links Arl8b and JIP4-Dynein complex to regulate lysosome size and positioningEditorial Note: Parts of this Peer Review File have been redacted as indicated to maintain the confidentiality of unpublished data.

REVIEWER COMMENTS

Reviewer #1 (Remarks to the Author):

Please find the review attached.

In their manuscript entitled “RUFY3 links Arl8b and JIP4-Dynein complex to regulate lysosome size and positioning”, Kumar et al. describe a novel dynein-dynactin adaptor for lysosomes that they name RUFY3, which is required for the perinuclear localisation of lysosomes. They employ several different techniques to show that RUFY3 is an Arl8b effector which is required for association of LAMP1+ve compartments with the dynein-dynactin complex, and for the maintenance of lysosome size and number. Overall, the study is exhaustive and carefully carried out and the manuscript is well-written. The primary concern that this reviewer has is the lack of proper biological role that can be attributed to RUFY3-mediated positioning and size of lysosomal compartments. The authors have made several observations including decrease in cell size upon RUFY3 overexpression, enhanced fission (and therefore decrease in size) of lysosomes upon RUFY3 depletion and aberrant dynamics of protein aggregate clearance in RUFY3 depleted cells. While all these are potentially interesting, it is unclear if any of these functions of RUFY3 are necessary for lysosomal function. The experiments required to make the conclusions in the manuscript are indicated in red below. If the authors are unable to perform these experiments due to COVID-19 lockdowns, this reviewer will be happy with changes in text to specify and/or tone down conclusions.

Major points:

1. In the introduction, the authors have equated Rab7+ve compartments to lysosomes. However, Rab7 endosomes are distinct organelles as documented in many publications. A proportion (~50%) co-localise with lysosomes, but Rab7 vesicles are not strictly degradative lysosomal organelles. In addition to delivering cargo to the lysosome, Rab7 compartments also play an important role in regulating endocytic cargo recycling in response to cellular stimuli and have been recently implicated in cargo secretion. Providing a distinction between Rab7 endosomes and lysosomes is biologically important.
2. Continuing from pt.1 above, there is lack of consistency in the manuscript with the marker employed for lysosomes – LAMP1, dextran and lysotracker have been interchangeably used to label lysosomes. Among the three, lysotracker would be only real lysosomal marker that needs to be employed in all the experiments to have consistent results and to draw appropriate conclusions for lysosomes.
3. Fig. 1 and 2: The RUFY3 subcellular localisation is inconsistent. It appears mostly cytosolic, with occasional puncta. An expanded panel as a supplementary figure showing the variation across experiments would be good, along with an associated quantification of cytosolic vs. punctate distribution. An explanation from the authors as to why they think this inconsistent pattern of RUFY3 localisation would be useful as well.
4. What is cytosolic leaching? A description about what cytosol leaching is, and how this is relevant to cellular physiology is required (if this of any relevance). Or was this performed to enhance the punctate localisation of RUFY3?
5. Fig. 1 N and O: The proportion of LAMP1-positive (or better, lysotracker +ve) compartments that localise with RUFY3 needs to be quantified to understand if the predominant adaptor on these compartments is RUFY3.
6. The positions of LAMP1/mitochondria etc. relative to the nucleus (“0-5um vs. >15um”) will be incorrect to compare between control and RUFY3 overexpressing cells due to reduction in the surface area of the cells in the latter. The positions need to be appropriately normalised to cell size.
7. Fig.4D-G and Fig. S4 A-D need to be quantified to back the authors’ conclusion that RUFY3 increases the association of dynactin and JIP4 with Arl8b.
8. Fig. 5: Since dextran is a cargo and not a lysosomal marker, experiments in Figure 5A-E need repeated using lysotracker.
9. Fig. S5A-D: Using two different colours of the same cargo (dextran) is not an appropriate method to determine trafficking into a compartment. Pre-loading the cells with dextran could prevent

further trafficking into the lysosomes meaning it is not possible to conclude that RUFY3 knockdown does not perturb cargo trafficking into the lysosome. As mentioned above, dextran is not a lysosomal marker, it is a cargo. Further, dextran is not a typical endocytic cargo. Dextran uptake is not receptor mediated and therefore does not give a broad representation of what is happening in terms of endocytic trafficking in the cell. For meaningful conclusions to be drawn from these experiments, they need to be repeated by incubating dextran and LDL (a well-established, receptor-mediated cargo that undergoes lysosomal degradation) in lysotracker-treated cells (control and RUFY3 knockdown). Given the importance of these experiments in understanding how RUFY3 ultimately regulates lysosomal function, these seem important.

10. Fig7 A-H: From these images, it appears that RUFY3 knockdown does not fully disrupt lysosomal positioning, as would be expected if it is the true adaptor required for dynein-dependent perinuclear lysosomal localisation. Further, LAMP1 labels a heterogenous population of lysosomes and lysosome-related organelles (<https://link.springer.com/article/10.1007/s00418-019-01842-z>, <https://rupress.org/jcb/article-standard/217/9/3127/120828/Characterization-of-LAMP1-labeled-nondegradative>). These experiments need to be repeated with lysotracker to confirm the effects RUFY3, JIP4 and dynein knockdown have on lysosomal positioning. Further, including Rab7 in these experiments would greatly strengthen the conclusions the authors are able to draw. There is no doubt RUFY3 is an adaptor between lysosomes and dynein, but it is clearly only one piece of the puzzle. By including Rab7, the authors would be able to determine with greater precision where RUFY3 acts. Based on their results it seems that may be important for recruiting dynein at the terminal step of late-endosome to lysosome transport; so, performing these experiments with Rab7 should help clarify this point.
11. Fig. 7I – J: These experiments need to be performed using lysotracker in control and RUFY3 knockdown cells. Again, the authors including Rab7 endosomes in the same basket as lysosomes does not entirely make sense. These images are very reminiscent of recently published data showing membrane buds off Rab7 endosomes to be re-delivered to Rab5 early endosomes (<https://journals.biologists.com/jcs/article/134/8/jcs254185/237792/De-novo-formation-of-early-endosomes-during-Rab5>). To claim that the tubulation and fission overserved here leads to formation of another lysosomal compartment and isn't rather part of the membrane flux between early and late endosomes, imaging Rab7 in similar conditions in control and RUFY3 knockdown cells is required.
12. Fig. S6: EGFR is a lysosomal targeted cargo in many, but not all circumstances. Conceptually, it is hard to reconcile how RUFY3 knockdown could lead to more degradation of EGFR, unless the level of EGF used induces an atypical cellular response such as macropinocytosis, which at 100ng per mL is likely. In Figure S5E-F it was indicated the RUFY3 knockdown could lead to perturbed proteolytic function of some lysosomes (although without this figure being quantified this cannot be concluded). Does RUFY3 knockdown lead to more proteolytically active lysosomes? Or is this a consequence of the often variable trafficking of EGFR? The above experiments should be repeated using LDL, as this will resolve how RUFY3 regulates degradation of a consistently lysosomal degraded cargo as opposed to a cargo such as EGFR that can be highly variable in its endocytic sorting.
13. Fig. 8H-M: The autophagy viewpoint is hard to follow. RUFY3 knockdown has little/no effect on LC3-positive compartments, but does have an effect (albeit modest) on the autophagy adaptor p62. The authors themselves query if these compartments are membrane encased, and this should be resolved. If it is ultimately the RUFY3 regulated positioning of lysosomes that is important for the degradation of a specific subset of autophagic cargo, then it should be demonstrated that this cargo is in the RUFY3 regulated lysosomes. The lysosomal encapsulation

of p62/Ub should be demonstrated with lysotracker and LAMP1 in control and RUFY3 knockdown cells.

14. Fig. 7 and 9: It is unclear how the authors explain the increased fission in RUFY3 depleted cells i.e., in the absence of RUFY3-mediated dynein association with lysosomes. Tug-of-war events which presumably are required for membrane fission necessarily require opposing forces on the same compartment by kinesin and dynein motors. In the absence of dynein-mediated forces (DHC RNAi), we would expect mislocalisation of lysosomes at the cell periphery due to a dominance of kinesin-mediated forces. The authors see this in Fig. 7D. If RUFY3 were the only cargo adaptor required for the perinuclear lysosomal positioning, we would expect to see a similar phenotype in RUFY3 and JIP4 depleted cells. However, Fig. 7B and C show a more homogenous distribution of LAMP1-positive cells, distinctly different from that of DHC RNAi. This likely indicates the role of other cargo adaptors – RILP/TMEM55 etc. in the positioning of these LAMP1-positive compartments.
15. Fig. 6K-N: The quantification of number of lysosomes needs to be normalised to the surface area of the cell that is visible in a micrograph.

Minor points

1. Fig. S2: EEA1 is a heterogeneous compartment that often co-localises variably with Rab5, Rab7 and indeed LAMP1 (<https://rupress.org/jcb/article-standard/216/10/3307/38957/WDR91-is-a-Rab7-effector-required-for-neuronal>, <https://rupress.org/jcb/article/217/9/3141/120859/Degradation-of-dendritic-cargos-requires-Rab7>). EEA1 therefore does not represent a *bona fide* early endosomal marker. Colocalisation of RUFY3 with Rab5 is therefore required to conclude that RUFY3 is not present on early endosomes. Similarly, transferrin is not a strict marker of recycling endosomes. While it does recycle *via* a Rab11 pathway, it can also be present in Rab5-positive early endosomes and EEA1 positive endosomes. To conclude that RUFY3 is not present on recycling endosomes, colocalisation of RUFY3 with Rab11 needs to be checked.
2. Fig. 3D – the signal of LAMP1 is barely visible.
3. Fig. 3H: The schematic seems to suggest that RUFY3 on its own associates with microtubules upon addition of rapamycin. This needs to be amended.
4. Fig. 5A-C: The movement will be clearer when the time lapses are represented as kymographs.
5. It would be good to increase the contrast in Fig. S5B and C (merged images).
6. Fig. S5E-F: These images require quantification. While the peripheral LAMP1-positive structures do indeed appear less Magic-Red positive in RUFY3 knockdown, the overall MagicRed intensity looks very similar between control and knockdown.
7. Discussion: expanding upon other late endosome/lysosome adaptors would help the authors support their conclusion. Specifically, expanding upon the function of RILP and how interaction of RILP and Rab7 regulates the balance of cargo exit from Rab7 endosomes before they become terminal lysosomes (<https://www.nature.com/articles/s41467-019-09437-x>), and comparing this to RUFY3 (which could be the adaptor that ensures Rab7 endosomes become the terminal lysosomes) would be useful.

Reviewer #2 (Remarks to the Author):

Kummar et al have identified a new effector (RUFY3) of the Arl8 small GTPase, which promotes the perinuclear localization of lysosomes by recruiting the dynein-dynactin adaptor protein JIP4. The authors demonstrate that RUFY3 is also critical for lysosome size and acidification, and perinuclear clustering of lysosomes in response to nutrient starvation.

The findings are novel and the authors have rigorously and thoroughly investigated RUFY3 in nearly all aspects of lysosome traffic, homeostasis and function in metabolic stress-induced and selective autophagy. There are no major concerns or issues with the paper's data and approaches - though there are a number points that should be addressed as outlined below.

Conceptually, however, the manuscript has a major weakness in neglecting to address previously established links between Arl8 and JIP4 with kinesin. Moreover, it does not address whether RUFY3 is a Rab7-independent mechanism of dynein recruitment. Some biochemical evidence for whether this mechanism is Rab7 independent and whether RUFY3 mediates a switch from the SKIP-mediated kinesin-driven transport to dynein transport is warranted. Is RUFY3/JIP4 bound to SKIP and kinesin, or is the SKIP-kinesin axis of Arl8-binding completely independent of the RUFY3-JIP4-dynein binding? Answers to these questions could be provided through co-IPs and western blots.

Specific points that require revisions:

1. In Figure 4C, the amount of RUFY3 co-IPing with JIP4 is very little. Quantification is necessary to determine whether this is above negative control levels. In addition, some explanation is needed for why this interaction is much weaker from the reverse co-IP of RUFY3 in figure 4B.

2. The Co-IP in Figure 4F suggests that Arl8 is required for the RUFY3-JIP4 interaction, but clearly RUFY3 and JIP4 co-IP at native levels of Arl8 as shown in Figure 4B. The correct experiment here is to perform a quantitative blot and compare the amounts of JIP4-RUFY3 association from steady state lysates and lysates of cells expressing Arl8-HA. Quantification should demonstrate the increase in the interaction in cells that express Arl8.

3. In Figure 5F, the decrease in the levels of DIC in the lysosomal fractions are not convincing. The amounts look comparable across the fractions of 1-3. Quantifications are required. Levels of DIC should be normalized to LAMP1 intensity and derived as percentage of total of DIC band intensities across the entire gradient for the individual fractions or for the fractions 1-3 or 1-4 combined. Clearly there is a shift in the knock-down with more lysosomes found in fraction 4, so this should be accounted in the quantifications. In addition, the Arl8 blot in this panel is not publication-quality.

4. Figure 5G: The GFP fluorescence is very low and not visible in these panels. GFP-RILP and -TMEM55B appear as smudges, and unclear if they localize to lysosomes. This needs to be improved.

5. Figure 6. The quantifications that report changes in lysosome size are difficult to trust as reliable. Particularly, in the case of the rescue experiment. The perinuclear clustering of lysosomes makes it nearly impossible to distinguish individual lysosomes in the perinuclear area in order to accurately measure their size. This is a major caveat and it's unclear how the authors were able to derive faithful quantifications on lysosome size from perinuclear areas.

6. The relationship between tubulation and fission events is unclear, and it's hard to follow the logic and data. Perhaps, it would be best to quantify tubulation events separately from fission events. There is no demonstration of actual fission in figure - a lysosome being split into two daughter lysosomes. Also, the presumption is that kinesin is mediating the tubulation of lysosomes by pulling membrane along microtubules, but why wouldn't dynein perform the same function? The authors should be also aware of data implicating JIP4 in actin-mediated (WASH) mechanisms of endosomal tubulation.

The authors should make an effort to improve conceptually this part of the manuscript. This is a bit weak as it's also unclear how the tubulation/fission relates to the acidification aspects. The latter is part of the lysosome maturation process, and the data imply that retrograde movement of lysosomes is required for lysosome maturation in both size and acidification. But this is not clearly communicated, if indeed this is what authors think it's happening.

7. It is unclear how the selective autophagy data (i.e., puromycin-induced aggregates) relate to the RUFY3-mediated retrograde movement of lysosomes. This part of the manuscript seems to be stretching far and beyond the main focus. I recommend to remove it as it does not appear to be directly relevant, and in its place, address some obvious questions about Arl8 and RUFY3 with respect to kinesin interaction and Rab7 GTPase. It is critical to provide some clarity on RUFY3 with respect to known interactions of its binding partners, and whether these are mutually exclusive or all in the same complex (see above).

Minor Comment:

- Please, add kD next to every number that corresponds to molecular weight (MW) markers in gel data and add the indication "MW" on top.

Reviewer #3 (Remarks to the Author):

Kumar et al. present a study on RUFY3, a protein identified as an Arl8B interactor. They identify two residues in RUFY3 required for Arl8B binding and find that co-expression of RUFY3 with Arl8B drives lysosomes to the perinuclear region. Via immunoprecipitation-mass spectrometry of RUFY3, they identify JIP4, dynein, and dynactin components as associated with RUFY3, and show that RUFY3, JIP4, and dynein are required for perinuclear distribution of lysosomes and regulate lysosome size. Together, their data suggests a model by which RUFY3 recruits dynein to lysosomes via an interaction with JIP4.

Overall, this is a very thorough, well-written paper identifying the novel adaptor RUFY3 and its mode of linking dynein to lysosomes. In most cases, the work strongly supports the claims made and makes a substantial contribution to the field. Overall, I recommend publication with the following revisions:

Major edits:

Figure 1F is a bit hard to follow. Figure 1F should have labels clearly defining which GST-RUFY3 protein is conjugated to beads in each lane, in addition to the arrows denoting size to the right. The ponceau stain gel is also hard to see. The WT GST-RUFY3 prep appears to have pretty low expression and isn't super clean. Ideally, the authors would optimize expression of GST-RUFY3 and clean up the prep. However, this data combined with their other data makes a pretty compelling case for a direct interaction, so at the very least, the

authors should perform a western blot to show alongside the Ponceau stain gel with a probe for GST or RUFY3 to ensure that RUFY3 is being expressed and to identify which band is RUFY3 in the WT condition (and potentially determine whether the other bands are degradation products). Finally, I couldn't find details on conditions for the GST-RUFY3 pulldown of His-Arl8B in the methods, so these should be added.

The findings in Figure 5F do not appear to support the claim made that DIC level is reduced in lysosomal fractions following RUFY3 depletion. To my eye, DIC levels do not appear to be reduced in the lysosomal fractions (highest LAMP1 signal in fractions ~1-4), but appear to be reduced in later fractions. This finding is also a bit hard to follow as DIC association does not appear to correspond with LAMP1 presence to begin with, making a further "loss" hard to identify. Clarification of this assay, better labeling and description, and quantification of multiple replicates is needed.

Does RUFY3 directly bind JIP4? Or are there other suspected intermediates between these proteins? In the discussion, the authors claim that RUFY3 "joins the league of other late endosomal/lysosomal proteins, including RILP, TRPML1, TMEM55B and SEPT9, which interact with dynein-dynactin retrograde motor either directly or via binding to dynein adaptors JIP3 or JIP4", but they never show that RUFY3 directly binds JIP4. I suggest that the authors perform this experiment with recombinant proteins or be clearer on this point in the results and discussion.

Is kinesin-1 still present on the perinuclear lysosomes containing RUFY3 and dynein? Knowing that could get at the mechanism of Arl8b-mediated dynein/kinesin competition. Are both dynein and kinesin present and associated with Arl8b on the lysosomes and one wins out depending on the circumstance? Or are they recruited/kicked off depending on context/presence or absence of RUFY3? The authors could either perform this experiment or postulate further in the discussion.

Minor edits:

In all cases where asterisks are used to mark transfected cells, it would be useful to also include asterisks or another denotation in the merged images.

In Figure 7I and J, it would be useful to have arrows denoting fission events.

The paragraph concerning Figure 8K-M is a bit hard to read. It seems the idea for Figure 8K-M is that in RUFY3-depleted cells, there is a delay in the timing of the normal aggregate formation and clearance process, but this idea isn't super formulated until the last sentence of the paragraph. It might be useful to have a sentence describing the standard timing of the aggregation formation/clearance process at the beginning of this paragraph, and then use that as a reference to describe the different defects seen in RUFY3-depleted cells.

Page 12: "drives lysosomes accumulation near the plasma membrane (see inset, Fig. 2F)". -> "drives lysosome accumulation" or "drives accumulation of lysosomes"

Page 16:

"These conclusions led to a hypothesis that RUFY3 recruits dynein motor on lysosomes and thereby mediate dynein-dependent lysosomal perinuclear positioning." 

"These conclusions led to a hypothesis that RUFY3 recruits the dynein motor on lysosomes and thereby mediates dynein-dependent lysosomal perinuclear positioning."

Page 17: "Previous studies have shown that the perinuclear and the peripheral pools of lysosomes have few differential characteristics and functions wherein the peripheral pool of

lysosomes is more poised for crosstalk and fusion with the plasma membrane and serum-dependent-mTORC1 activation (Jia and Bonifacino, 2019; Korolchuk et al., 2011; Pu et al., 2017)." 

"Previous studies have shown that the perinuclear and the peripheral pools of lysosomes have few differential characteristics and functions. The peripheral pool of lysosomes is more poised for crosstalk and fusion with the plasma membrane and serum-dependent-mTORC1 activation (Jia and Bonifacino, 2019; Korolchuk et al., 2011; Pu et al., 2017)."

Page 22: "This was in contrast to the control siRNA treated cells, where as expected, lysosomes were accumulated in the perinuclear region in all three conditions of starvation"  I think precise wording here is important as many RUFY3 depleted cells appear to have some lysosomal accumulation in both the perinuclear and peripheral region. I would suggest slightly modifying this sentence to "This was in contrast to the control siRNA treated cells, where as expected, lysosomes were accumulated in the perinuclear region and generally absent from the periphery in all three conditions of starvation" as both seem to be true, and the peripheral localization appears to be the most stark when looking at the images.

Page 24: "In contrast, we found a striking decrease in the formation of these aggregated punctae upon RUFY3 depletion (lower panel, Fig. 8K; quantification is shown in Fig. 8M)." Minor, but to me this is one of the least striking results of the whole paper. I suggest toning down the language a bit.

Page 27: "A third reason could be that while markers like LAMP1 are common, but different dynein adaptors are essentially required for the motility of distinct compartments (Fig. 9A (III))."  "A third reason could be that while markers like LAMP1 are common, different dynein adaptors are essentially required for the motility of distinct compartments (Fig. 9A (III))."

Should cite recent Cason et al. 2021 paper "Sequential dynein effectors regulate axonal autophagosome motility in maturation-dependent pathway".

Reviewer #4 (Remarks to the Author):

I was asked to review the mass spectrometry portion of this paper. Unfortunately, the information provided for this portion of the work is incomplete. Absolutely no details were provided by the authors on any aspect of the mass spectrometry analysis or the analysis of the mass spectrometry data (database searching). It is also standard practice to upload raw data files to a public repository such as ProteomeXchange. Until the authors provide the same level of detail for the mass spectrometry portion of this work as they do for other experiments, this work is not suitable for publication. The authors are urged to consult with the proteomics core that ran their samples in order to get the necessary details.

Response to reviewer comments:

We are grateful to the reviewers of the previously submitted manuscript for helpful criticisms and were heartened by their positive comments on the strength of the study. We have included below: (i) a list outlining the new experiments added to the manuscript and (ii) a detailed point-by-point response to each reviewer's comments.

(i) New experimental data added to the manuscript

1. In response to reviewer #1 (points #2, #8 and #10), we have employed LysoTracker as the probe for lysosomes and quantified lysosome positioning in RUFY3, JIP4 and dynein knockdown cells. Data with LysoTracker show a similar phenotype as LAMP1, namely, that a subset of late endosomes/lysosomes (that are primarily Arl8b-positive) are localized to the cell periphery in RUFY3-depleted cells and mobile population of lysosomes is increased upon RUFY3 knockdown. As expected, JIP4 and dynein knockdown showed a more profound effect on the peripheral positioning of lysosomes than RUFY3 silencing owing to their multiple interaction partners, which regulate lysosomal motility. These results are described in the revised text (**lines #273-287 and #388-404**) and presented in the new panels **Fig. 3A-F, Fig. 6A-E and Supplementary Fig. S4E-I**.

2. In response to reviewer #1 (point #3), we have added a panel of images showing expression-dependent localization of epitope-tagged-RUFY3 (**Supplementary Fig. S2A-C; lines #187-196**). Overall, the data suggest that limiting expression of endogenous Arl8b could be the primary reason for cytosolic RUFY3-HA distribution.

3. In response to reviewer #1 (point #5), we have added Mander's coefficient of overlap of LAMP1 with RUFY3 (**Fig. 1Q; lines #214-230**). This analysis shows that RUFY3 localizes to only a select population of lysosomes, which our new data suggests should be Arl8b-positive compartments. In this revised manuscript, we have added several lines of evidence to clarify where RUFY3 acts in the endolysosomal pathway. Overall, this data indicates that, unlike Arl8b, RUFY3 does not interact with Rab7 and RUFY3-mediated perinuclear lysosome positioning is independent of Rab7 (**Fig. 4; lines #319-357**).

4. In response to reviewer #1 (point #6), we have quantified organelle positioning by measuring fractional distances corresponding to the (background-corrected) organelle marker fluorescence intensities (LAMP1/LysoTracker/SiR-Lysosome and TOM-20 for Mitochondria) relative to the maximum distance from the center of the nucleus to the cell periphery. These results are described in the revised text (**lines #214-218; #273-287 and #302-315**) and presented in the new panels **Supplementary Fig. S1L-M, S2C and S2K**.

5. In response to reviewer #1 (point #7), we have quantified the images and blots to support our conclusion that RUFY3 increases the association of dynactin and JIP4 with Arl8b (**Fig. 5C-M; Supplementary Fig. S4A-C; lines #368-384**).

6. In response to reviewer #1 (points #9 and #12), we have quantified colocalization of LDL and dextran with LysoTracker in RUFY3-depleted cells (**Supplementary Fig. S5**). Our findings indicate that RUFY3 does not regulate the delivery of endocytic cargo such as LDL and dextran to lysosomes. It is important to note that LysoTracker also defines

acidic late endosomes (positive for Rab7) in addition to endolysosomes. Interestingly, we did observe a modest decrease (~25%) in BODIPY-BSA fluorescence in RUFY3-depleted cells (**Fig. 8K-M; lines #529-549**). As dequenching of BODIPY-BSA fluorescence is observed upon its proteolytic cleavage in lysosomes, these results indicate that lysosomes might be less degradative in RUFY3-depleted cells. The impaired degradative ability could be due to reduced lysosome size, as shown in a previous study where lysosomal cargo degradation was less in cells with decreased lysosome size (Meneses-Salas et al., 2020). Furthermore, as Rab7 and Arl8b colocalization is modestly reduced in RUFY3-depleted cells (**Fig. 4M**), endolysosome formation (generally regarded as the degradative compartments) might be reduced in RUFY3 knockdown.

7. In response to reviewer #1 (point #10) and reviewer #2 comments, we have added new experiments that suggest RUFY3 is a dynein adaptor for Arl8b-positive compartments (representing likely endolysosomes and terminal (storage) lysosomes). Also, as the reviewer predicted, we find a reduced overlap between Rab7 and Arl8b in RUFY3-depleted cells (**Fig. 4; lines #319-357**).

8. In response to reviewer #2 (point #3) and reviewer #3 (point #2) comments, we have quantified DIC levels in the lysosomal fraction in RUFY3 depletion. In agreement with our hypothesis that RUFY3 is one of the dynein adaptors on lysosomes, DIC levels were reduced in lysosomal fractions upon RUFY3 depletion (**Fig. 6F-G; lines #404-412**).

9. In response to reviewer #2 (point #1 and #2) and reviewer #3 (point #3) comments, we have quantified the co-immunoprecipitation blots showing that RUFY3 is the linker between JIP4 and Arl8b (**Fig. 5G-M; Supplementary Fig. S4A-C and lines #375-383**). Further, we have demonstrated the interaction between JIP4 (immunopurified from mammalian cells) and recombinant GST-RUFY3. Our data strongly hints at a direct interaction between RUFY3 and JIP4. These results are described in the new panels **Fig. 5A-B (lines #368-374)**.

(ii) Detailed point-by-point response to reviewers comments

In the following paragraphs we have address the questions and criticisms of the reviewers.

Response to Reviewer #1 Comments:

In their manuscript entitled “RUFY3 links Arl8b and JIP4-Dynein complex to regulate lysosome size and positioning”, Kumar et al. describe a novel dynein-dynactin adaptor for lysosomes that they name RUFY3, which is required for the perinuclear localisation of lysosomes. They employ several different techniques to show that RUFY3 is an Arl8b effector which is required for association of LAMP1+ve compartments with the dynein-dynactin complex, and for the maintenance of lysosome size and number. Overall, the study is exhaustive and carefully carried out and the manuscript is well-written. The primary concern that this reviewer has is the lack of proper biological role that can be attributed to RUFY3-mediated positioning and size of lysosomal compartments. The authors have made several observations including decrease in cell size upon RUFY3 overexpression, enhanced fission (and therefore decrease in size) of lysosomes upon RUFY3 depletion and aberrant dynamics of protein aggregate clearance in RUFY3-depleted cells. While all these are potentially interesting, it is unclear if any of these

functions of RUFY3 are necessary for lysosomal function. The experiments required to make the conclusions in the manuscript are indicated in red below. If the authors are unable to perform these experiments due to COVID-19 lockdowns, this reviewer will be happy with changes in text to specify and/or tone down conclusions.

We thank the reviewer for appreciating our research work and for raising important concerns that have allowed us to improve this study. Please find our response to comments from reviewer #1.

Major points:

1. In the introduction, the authors have equated Rab7+ve compartments to lysosomes. However, Rab7 endosomes are distinct organelles as documented in many publications. A proportion (~50%) co-localise with lysosomes, but Rab7 vesicles are not strictly degradative lysosomal organelles. In addition to delivering cargo to the lysosome, Rab7 compartments also play an important role in regulating endocytic cargo recycling in response to cellular stimuli and have been recently implicated in cargo secretion. Providing a distinction between Rab7 endosomes and lysosomes is biologically important.

Response: We agree with the reviewer, and we have modified the introduction text and replaced the word lysosomes with late endocytic compartments that better describe the Rab7-positive compartments (**lines # 68-73**). Indeed, Rab7-RILP and Rab7-FYCO1 complexes are well accepted for their role as motor-adaptor complexes on the late endosomes (Ballabio A and Bonifacino J Nat Rev Mol Cell Biol. 2020; Neefjes J et al. Trends in Cell Biol. 2017; Pu J et al. J Cell Sci. 2016). Upon RUFY3 depletion, Rab7-positive endosomes are modestly shifted to the cell periphery, while Arl8b endosomes show a striking shift to the cell periphery (**Fig. 4J- M**). We also find that RUFY3 does not interact with Rab7 (**Fig. 4A-B**), and neither Rab7 is required for RUFY3 membrane localization (**Fig. 4C-I**). Overall, our data agree with the hypothesis that RUFY3 is a specific dynein adaptor for Arl8b-positive late endocytic compartments. We think these compartments are likely to be terminal lysosomes and endolysosomes based on the prior literature.

2. Continuing from pt.1 above, there is lack of consistency in the manuscript with the marker employed for lysosomes – LAMP1, dextran and lysotracker have been interchangeably used to label lysosomes. Among the three, lysotracker would be only real lysosomal marker that needs to be employed in all the experiments to have consistent results and to draw appropriate conclusions for lysosomes.

Response: To address this concern, besides LAMP1, we have now added quantification of lysosome positioning using Lysotracker and SiR-Lysosome (a fluorescent probe based on the pepstatin A, a cathepsin D binding peptide) in **Fig. 3**. We find that all three markers/probes-LAMP1, Lysotracker and SiR-Lysosome show a similar peripheral positioned population upon RUFY3 depletion (**Fig. 3A-F**). In experiments shown in Figs. 1 and 2, we have retained LAMP1 as the lysosomal marker because here, we had to fix and permeabilize the cells for immunostaining RUFY3. We find that Lysotracker and SiR-Lysosome signal is very weak (almost undetectable) after permeabilization of cells (please see a representative image below of Lysotracker and SiR-Lysosome before and after permeabilization; **Fig. A**).

Figure A

Fig. A: Permeabilization of cells leads to loss of Lysotracker and SiR-Lysosome fluorescence signal. HeLa cells were labeled with Lysotracker or SiR-Lysosome probe for 1 hr. Post labeling, cells were fixed with 4% paraformaldehyde (*left panels*) or fixed and permeabilized with 0.2% saponin (*right panels*). As can be seen in confocal micrographs, permeabilization of cells leads to loss of Lysotracker and SiR-Lysosome fluorescence signal. Scale Bars: 10 μ m

3. Fig. 1 and 2: The RUFY3 subcellular localisation is inconsistent. It appears mostly cytosolic, with occasional punctae. An expanded panel as a supplementary figure showing the variation across experiments would be good, along with an associated quantification of cytosolic vs. punctate distribution. An explanation from the authors as to why they think this inconsistent pattern of RUFY3 localisation would be useful as well.

Response: We thank the reviewer for this suggestion. To clarify this point, we have now added new image panels showing our observations with the subcellular localization of epitope-tagged RUFY3 (**Supplementary Fig. S2A-C**). Cells with moderate to high expression of RUFY3 show a cytosolic localization of RUFY3-HA, while in <20% of cells with weak to moderate expression, we find a cytosolic distribution and few punctae. To better depict RUFY3 colocalization with LAMP1, we have briefly permeabilized the cells before fixation (**Fig. 1I**). As shown in **Fig. 1M**, co-expression of Arl8b with RUFY3 significantly promotes its membrane/lysosomal recruitment. Thus, limiting expression of endogenous Arl8b could be the primary reason for cytosolic RUFY3-HA distribution. This information is now added in the manuscript text (**lines # 187-196**).

4. What is cytosolic leaching? A description about what cytosol leaching is, and how this is relevant to cellular physiology is required (if this of any relevance). Or was this performed to enhance the punctate localisation of RUFY3?

Response: To perform “cytosol leaching”, cells were permeabilized (0.05% Saponin, 5 minutes incubation on ice) before fixation. There are several references where this strategy has been employed to observe membrane-bound localization of a protein when it is masked by the abundant cytosolic expression (Pedersen NM et al. J Cell Biol. 2020; Liese S et al. Proc Natl Acad Sci USA. 2020; Wenzel EM et al. Nat Commun. 2018; Hong Z et al. J Cell Biol. 2017). We have now included the protocol for performing cytosol leaching in the “immunostaining” subheading of the materials & method section (**lines # 1155-1159**). Our claim that RUFY3 punctae colocalize with lysosomal markers in cytosol-leached cells supports that RUFY3 is present on lysosomes under physiological conditions (**Fig. 1I** and **Supplementary Fig. S2J**). This claim is supported by several other independent lines of evidence:

- 1) Endogenous RUFY3 is present in LYSO-IP (**Fig. 1K**).
- 2) RUFY3 is an effector of lysosomal G protein Arl8b (**Fig. 1B-D**).
- 3) RUFY3 expression regulates lysosomal positioning (**Figs. 2 and 3**).

5. **Fig. 1 N and O: The proportion of LAMP1-positive (or better, lysotracker +ve) compartments that localise with RUFY3 needs to be quantified to understand if the predominant adaptor on these compartments is RUFY3.**

Response: We agree with the reviewer’s suggestion. We have now included Mander’s coefficient of LAMP1 overlap with RUFY3, which shows an average value of 0.22. Upon co-expression with Arl8b, we find the average value of LAMP1 overlap with RUFY3 increases to 0.44 (**Fig. 1Q**). Indeed, this suggests that RUFY3 may localize to only a subset of late endosomes/lysosomes.

6. The positions of LAMP1/mitochondria etc. relative to the nucleus (“0-5um vs. >15um”) will be incorrect to compare between control and RUFY3 overexpressing cells due to reduction in the surface area of the cells in the latter. The positions need to be appropriately normalised to cell size.

Response: To address this concern, we have now quantified organelle positioning by measuring “fractional distances” as described previously (Guardia CM et al. Cell Rep. 2016; Jongsma ML et al. Cell. 2016). The mean distances corresponding to the (*background-corrected*) organelle marker fluorescence intensities (LAMP1 and TOM-20 for mitochondria) relative to the maximum distance from the center of the nucleus to the cell periphery were reported as fractional distances in dot-plot graphs (**Supplementary Figs. S2L-M, S3C and S3K**). A description of this method is included in the materials and method section (**lines # 1204-1216**). Further, we want to mention that binucleated cells and non-uniform shaped cells were not included for the quantification of lysosome positioning.

7. **Fig.4D-G and Fig. S4 A-D need to be quantified to back the authors’ conclusion that RUFY3 increases the association of dynactin and JIP4 with Arl8b.**

Response: We appreciate the reviewer’s criticism, and accordingly, we have quantified the data shown in these representative images. The quantification is now shown in **Fig. 5I, Fig. 5K, Fig. 5M** and **Supplementary Fig. S4C** along with the representative images. Our results support that RUFY3 links the Arl8b and JIP4-dynein-dynactin complex.

8. Fig. 5: Since dextran is a cargo and not a lysosomal marker, experiments in Figure 5A-E need repeated using lysotracker.

Response: We thank the reviewer for this comment. As suggested by the reviewer, we have now checked the motility behavior of lysosomes upon RUFY3 depletion by using Lysotracker. These new experiments' results agree with our previous findings that RUFY3 silencing significantly increased the total mobile fraction and average speed of lysosomes (**Fig. 6A-E** in the revised manuscript).

9. Fig. S5A-D: Using two different colours of the same cargo (dextran) is not an appropriate method to determine trafficking into a compartment. Pre-loading the cells with dextran could prevent further trafficking into the lysosomes meaning it is not possible to conclude that RUFY3 knockdown does not perturb cargo trafficking into the lysosome. As mentioned above, dextran is not a lysosomal marker, it is a cargo. Further, dextran is not a typical endocytic cargo. Dextranuptake is not receptor mediated and therefore does not give a broad representation of what is happening in terms of endocytic trafficking in the cell. For meaningful conclusions to be drawn from these experiments, they need to be repeated by incubating dextran and LDL (a well established, receptor-mediated cargo that undergoes lysosomal degradation) in lysotracker-treated cells (control and RUFY3 knockdown). Given the importance of these experiments in understanding how RUFY3 ultimately regulates lysosomal function, these seem important.

Response: We thank the reviewer for this important comment, and we have now performed the pulse-chase analysis of endocytosed cargo-LDL and dextran and measured their colocalization with Lysotracker at different time points. Our findings indicate that RUFY3 does not regulate the delivery of endocytic cargo such as LDL and dextran to lysosomes (**Supplementary Fig. S5**). Interestingly, we did observe a modest decrease (~25%) in BODIPY-BSA fluorescence in RUFY3-depleted cells. As dequenching of BODIPY-BSA fluorescence is observed upon its proteolytic cleavage in lysosomes, these results indicate that lysosomes might be less degradative in RUFY3-depleted cells (**Fig. 8K-M**).

10. Fig7 A-H: From these images, it appears that RUFY3 knockdown does not fully disrupt lysosomal positioning, as would be expected if it is the true adaptor required for dynein-dependent perinuclear lysosomal localisation. Further, LAMP1 labels a heterogenous population of lysosomes and lysosome-related organelles (<https://link.springer.com/article/10.1007/s00418-019-01842-z>, <https://rupress.org/jcb/article-standard/217/9/3127/120828/Characterization-of-LAMP1-labeled-nondegradative>). These experiments need to be repeated with lysotracker to confirm the effects RUFY3, JIP4 and dynein knockdown have on lysosomal positioning. Further, including Rab7 in these experiments would greatly strengthen the conclusions the authors are able to draw. There is no doubt RUFY3 is an adaptor between lysosomes and dynein, but it is clearly only one piece of the puzzle. By including Rab7, the authors would be able to determine with greater precision where RUFY3 acts. Based on their results it seems that may be important for recruiting dynein at the terminal step of late-endosome to lysosome transport; so, performing these experiments with Rab7 should help clarify this point.

Response: We thank the reviewer for these crucial insights. We have now employed Lysotracker to report the lysosomal positioning phenotype in RUFY3, JIP4 and

dynein depletion (**Supplementary Fig. S4E-I**). We agree that, unlike JIP4 or dynein, RUFY3 alters the localization of only a subset of lysosomes. Our findings presented in the new **Fig. 4J-K** shows that the peripheral lysosomal compartments are predominantly enriched for Arl8b and less for Rab7. Thus, the evidence suggests that RUFY3 is a dynein adaptor for Arl8b-positive compartments (likely representing endolysosomes and terminal (storage) lysosomes). Also, as the reviewer predicted, we find a reduced overlap between Rab7 and Arl8b in RUFY3 depleted cells (**Fig. 4M**).

11. **Fig. 7I – J:** These experiments need to be performed using lysotracker in control and RUFY3 knockdown cells. Again, the authors including Rab7 endosomes in the same basket as lysosomes does not entirely make sense. These images are very reminiscent of recently published data showing membrane buds off Rab7 endosomes to be re-delivered to Rab5 early endosomes (<https://journals.biologists.com/jcs/article/134/8/jcs254185/237792/De-novo-formation-of-early-endosomes-during-Rab5>). To claim that the tubulation and fission overserved here leads to formation of another lysosomal compartment and isn't rather part of the membrane flux between early and late endosomes, imaging Rab7 in similar conditions in control and RUFY3 knockdown cells is required.

Response: We have consistently observed a reduced lysosome area in RUFY3 depletion, as confirmed by direct visualization of lysosomes by electron microscopy. In our original submission, we had quantified the membrane budding events from LAMP1 vesicles in control and RUFY3 depletion. In these live videos, we could not observe a clear tubulation, followed by fission. Based on the reviewers 1 and 2 comments that tubulation and fission should be separately quantified and imaging of Rab7 should be performed, we tried several new experiments, including imaging of Arl8b-GFP/GFP-Rab7/LAMP1-GFP along with SiR-Lysosome/Lysotracker to observe events that show a distinct tubulation and subsequent fission of the late endocytic compartments. We could not obtain data showing tubulation of late endosomes/lysosomes and then fission from the tubule in these experiments imaged by conventional confocal laser scanning microscope. This issue could be due to the limitations of the speed of imaging and that lysosome tubulation, and fission events are difficult to capture in HeLa cells compared to autolysosome reformation events where longer tubulation is observed. Indeed, we were able to capture videos showing LAMP1 tubulation when we starved cells for 8 hr (**Fig. B**). Thus, we have chosen to tone down our claim of *how* lysosome size is reduced upon RUFY3 depletion and focus our revised study on clarifying the role of RUFY3 as a specific dynein adaptor for Arl8b-positive endosomes.

Figure B

Fig. B: Long-term nutrient starvation augments lysosome tubulation in cells. HeLa cells transfected with LAMP1-GFP were subjected to EBSS treatment for 8 hr. The time-lapse series was captured with a frame time of 0.97 sec, and representative micrographs depicting LAMP1-GFP-positive vesicles undergoing tubulation events are marked with yellow arrows. Scale Bars: 10 μm (main); 2 μm (inset).

12. Fig. S6: EGFR is a lysosomal targeted cargo in many, but not all circumstances. Conceptually, it is hard to reconcile how RUFY3 knockdown could lead to more degradation of EGFR, unless the level of EGF used induces an atypical cellular response such as macropinocytosis, which at 100ng per mL is likely. In Figure S5E-F it was indicated the RUFY3 knockdown could lead to perturbed proteolytic function of some lysosomes (although without this figure being quantified this cannot be concluded). Does RUFY3 knockdown lead to more proteolytically active lysosomes? Or is this a consequence of the often variable trafficking of EGFR? The above experiments should be repeated using LDL, as this will resolve how RUFY3 regulates degradation of a consistently lysosomal degraded cargo as opposed to a cargo such as EGFR that can be highly variable in its endocytic sorting.

Response: We thank the reviewer for this important comment. We have now replaced EGFR data with the pulse-chase analysis of endocytic cargo- LDL and dextran and measured their colocalization with LysoTracker at different time points. Our findings indicate that RUFY3 does not regulate the delivery of endocytic cargo (such as LDL and dextran) to LysoTracker-positive vesicles (**Supplementary Fig. S5**). Interestingly, we did observe a modest (~25%) decrease in BODIPY-BSA fluorescence in RUFY3-depleted cells (**Fig. 8K-M**). As dequenching of BODIPY-BSA fluorescence is observed upon its proteolytic cleavage in lysosomes, these results indicate that lysosomes might be less degradative in RUFY3-depleted cells. The decrease in degradative ability could be due to reduced lysosome size, as shown in a previous study where lysosomal cargo degradation was less in cells with decreased lysosome size (Meneses-Salas E et al. Cells. 2020). Additionally, colocalization of Rab7 and Arl8b is reduced in RUFY3-depleted cells, suggesting that there might be reduced fusion of Rab7 and Arl8b vesicles (generally regarded as the degradative endolysosomal compartment) in RUFY3 knockdown (**Fig. 4M**). Nevertheless, we also performed EGFR degradation at a lower concentration of EGF, i.e. 20 ng/mL (**Fig. C**). Similar to 100 ng/mL EGF concentration, we also observed an increase in EGFR degradation in RUFY3-depleted cells. We are currently investigating why EGFR degradation is enhanced while degradation of luminal cargo is reduced upon RUFY3 knockdown. Interestingly, PLEKHM1 depletion also enhances EGFR degradation, while degradation of luminal cargo like DQ-BSA is reduced (Tabata K et al. Mol Biol Cell. 2010; Marwaha R et al. J Cell Biol. 2017).

Figure C

Fig. C: EGFR degradation assay. Control and RUFY3-silenced HeLa cells were serum- starved for 1 hr and incubated with two different concentrations of EGF (100 ng/mL, *left panels* or 20 ng/mL, *right panels*) for the indicated time points. At the end of each time point, cell lysates were prepared and IB with indicated antibodies.

13. Fig. 8H-M: The autophagy viewpoint is hard to follow. RUFY3 knockdown has little/no effect on LC3-positive compartments, but does have an effect (albeit modest) on the autophagy adaptor p62. The authors themselves query if these compartments are membrane encased, and this should be resolved. If it is ultimately the RUFY3 regulated positioning of lysosomes that is important for the degradation of a specific subset of autophagic cargo, then it should be demonstrated that this cargo is in the RUFY3 regulated lysosomes. The lysosomal encapsulation of p62/Ub should be demonstrated with lysotracker and LAMP1 in control and RUFY3 knockdown cells.

Response: As suggested by both reviewers #1 and #2, we agree that selective autophagy is farther from the focus of this work. Thus, we have removed the puromycin-induced p62/Ub aggregate data in this revised version of the manuscript. Overall, we find that RUFY3 does not appear as a major player in vesicle fusion with lysosomes, and this includes autophagosome-lysosome fusion (**Fig. 8H-J**).

14. Fig. 7 and 9: It is unclear how the authors explain the increased fission in RUFY3 depleted cells i.e., in the absence of RUFY3-mediated dynein association with lysosomes. Tug-of-war events, which presumably are required for membrane fission necessarily, require opposing forces on the same compartment by kinesin and dynein motors. In the absence of dynein-mediated forces (DHC RNAi), we would expect mislocalisation of lysosomes at the cell periphery due to a dominance of kinesin-mediated forces. The authors see this in Fig. 7D. If RUFY3 were the only cargo adaptor required for the perinuclear lysosomal positioning, we would expect to see a similar phenotype in RUFY3 and JIP4 depleted cells. However, Fig. 7B and C show a more homogenous distribution of LAMP1-positive cells, distinctly different from that of DHC RNAi. This likely indicates the role of other cargo adaptors – RILP/TMEM55 etc. in the positioning of these LAMP1-positive compartments.

Response: We thank the reviewer for this insightful comment. Indeed dynein depletion has a striking effect on lysosome positioning, where the majority of lysosomes are shifted to the cell periphery (**Supplementary Fig. S4H**). In comparison, RUFY3-depleted cells clearly show only a subset of LysoTracker/SiR- Lysosome/LAMP1 compartments localized to the periphery (**Fig. 3A-C**). From our analysis that Rab7 does not interact with RUFY3 (**Fig. 4**), we propose that this subset of LysoTracker compartments should be Arl8b positive, while Rab7-only compartments likely retain their normal localization upon RUFY3 depletion. We agree with the reviewer that besides RUFY3, there are other dynein adaptors for lysosome perinuclear positioning. As we have shown in **Fig. 6H-K**, dynein adaptors such as RILP and TMEM55 are functional in RUFY3-depleted cells and, upon overexpression, can rescue lysosome positioning in RUFY3 depletion. In the discussion section, we have proposed various hypothetical scenarios for explaining the presence of multiple dynein adaptors on lysosomes (**Fig. 9A; lines #585-630**).

15. Fig. 6K-N: The quantification of number of lysosomes needs to be normalised to the surface area of the cell that is visible in a micrograph.

Response: We agree with the reviewer's comments and have now quantified the lysosome number normalized to the visible surface area of the cell (**Fig. 7G-J**).

Minor points:

1. Fig. S2: EEA1 is a heterogeneous compartment that often co-localises variably with Rab5, Rab7 and indeed LAMP1 (<https://rupress.org/jcb/article-standard/216/10/3307/38957/WDR91-is-a-Rab7-effector-required-for-neuronal>, <https://rupress.org/jcb/article/217/9/3141/120859/Degradation-of-dendritic-cargos-requires-Rab7>). EEA1 therefore does not represent a *bona fide* early endosomal marker. Colocalisation of RUFY3 with Rab5 is therefore required to conclude that RUFY3 is not present on early endosomes. Similarly, transferrin is not a strict marker of recycling endosomes. While it does recycle via a Rab11 pathway, it can also be present in Rab5-positive early endosomes and EEA1 positive endosomes. To conclude that RUFY3 is not present on recycling endosomes, colocalisation of RUFY3 with Rab11 needs to be checked.

Response: We thank the reviewer for this comment. We have included the quantification of RUFY3 colocalization with Rab5 (**Supplementary Fig. S2E**) and Rab11 (**Supplementary Fig. S2G**; quantification reported in **Fig. 1J** and **Supplementary Fig. S2H-I**) in the revised manuscript.

2. Fig. 3D – the signal of LAMP1 is barely visible.

Response: We have replaced the previous Fig. 3D with better representative images (now *third* and *fourth* panels in **Fig. 3A**).

3. Fig. 3H: The schematic seems to suggest that RUFY3 on its own associates with microtubules upon addition of rapamycin. This needs to be amended.

Response: We thank the reviewer for noticing this error. We have modified the schematic (**Fig. 3G**).

4. Fig. 5A-C: The movement will be clearer when the time lapses are represented as kymographs.

Response: We tried performing the analysis using kymographs but found the “trackmate” plugin better suited for HeLa cells. Also, there are several references in literature where use of “trackmate” plugin is reported for tracking of lysosomes (Jongsma ML et al. EMBO J. 2020; Jongsma ML et al. Cell. 2016; Wijdeven RH et al. Nat Commun. 2015).

5. It would be good to increase the contrast in Fig. S5B and C (merged images).

Response: As suggested by the reviewer (major point #9), we have now removed this data in the revised manuscript.

6. Fig. S5E-F: These images require quantification. While the peripheral LAMP1- positive structures do indeed appear less Magic-Red positive in RUFY3 knockdown, the overall MagicRed intensity looks very similar between control and knockdown.

Response: These data from our original submission referred to “lysosome fission” observed in RUFY3-depleted cells. We have now removed this data in the revised manuscript.

7. Discussion: expanding upon other late endosome/lysosome adaptors would help the authors support their conclusion. Specifically, expanding upon the function of RILP and how interaction of RILP and Rab7 regulates the balance of cargo exit from Rab7 endosomes before they become terminal lysosomes (<https://www.nature.com/articles/s41467-019-09437-x>), and comparing this to RUFY3 (which could be the adaptor that ensures Rab7 endosomes become the terminal lysosomes) would be useful.

Response: We thank the reviewer for these suggestions. We have included these points in the discussion (lines #613-630).

Response to Reviewer #2 Comments:

Kummar et al have identified a new effector (RUFY3) of the Arl8 small GTPase, which promotes the perinuclear localization of lysosomes by recruiting the dynein-dynactin adaptor protein JIP4. The authors demonstrate that RUFY3 is also critical for lysosome size and acidification, and perinuclear clustering of lysosomes in response to nutrient starvation. The findings are novel and the authors have rigorously and thoroughly investigated RUFY3 in nearly all aspects of lysosome traffic, homeostasis and function in metabolic stress-induced and selective autophagy. There are no major concerns or issues with the paper's data and approaches - though there are a number of points that should be addressed as outlined below.

Conceptually, however, the manuscript has a major weakness in neglecting to address previously established links between Arl8 and JIP4 with kinesin. Moreover, it does not address whether RUFY3 is a Rab7-independent mechanism of dynein recruitment. Some biochemical evidence for whether this mechanism is Rab7 independent and whether RUFY3 mediates a switch from the SKIP-mediated kinesin-driven transport to dynein transport is warranted. Is RUFY3/JIP4 bound to SKIP and kinesin, or is the SKIP- kinesin axis of Arl8-binding completely independent of the RUFY3-JIP4-dynein binding? Answers to these questions could be provided through co-IPs and western blots.

Response: We thank the reviewer for appreciating our research findings and the insightful comments. We have added new **Fig. 4**, where we have analyzed the role of Rab7 in regulating RUFY3 localization and function in lysosome positioning. In summary, these experiments clearly show that RUFY3 does not interact with Rab7 and its membrane localization and its effect on lysosome positioning is independent of Rab7. These results suggest that unlike other shared interaction partners of Rab7 and Arl8b, i.e. PLEKHM1 and SKIP/PLEKHM2, RUFY3 is a specific interactor of Arl8b.

We next proceeded with the reviewer's suggestion to analyze RUFY3 role in regulating Arl8b-SKIP-Kinesin-1 interaction and JIP4-Kinesin-1 interaction. Surprisingly, these results have revealed that RUFY3 promotes SKIP-Kinesin-1 interaction (**Fig. D-F**) with Arl8b but does not regulate JIP4-Kinesin-1 interaction (**Fig. G**). This is a consistent finding in our hands.

[Redacted]

Considering our major finding that RUFY3 is a dynein adaptor on lysosomes, how are these observations explained?

We hypothesize that RUFY3 promotes the recruitment of kinesin on Arl8b-positive compartments but that this kinesin is autoinhibited and unable to mediate the peripheral

motility of these compartments. Increasing SKIP expression relieves kinesin autoinhibition, as observed that lysosomes move towards the cell periphery in SKIP overexpressing cells (**Fig. H**). Thus, SKIP overcomes the RUFY3 effect on lysosome positioning and moves the lysosome to the periphery. Notably, RUFY3 continues to localize to these peripheral lysosomes. Moreover, SKIP does not alter RUFY3 binding to Arl8b (**Fig. I**).

We think these observations hint towards the role of RUFY3 as an adaptor that can bind to both dynein and kinesin, and probably RUFY3 promotes loading of inactive kinesin on lysosomes. Arl8b-binding to SKIP and SKIP recruitment then subsequently activates Kinesin-1. Indeed, some recent examples of adaptors, including Hook3, JIP3/4, TRAK2 that bind to both dynein and kinesin through distinct domains and might be involved in motor loading and recycling (Fenton AR et al. Nat. Commun. 2021; Cason SE et al. J Cell Biol. 2021; Kendrick AA et al. J Cell Biol. 2019; Willett R et al. Nat. Commun. 2017; Sato T et al. Cell Death Differ. 2015; Drerup CM and Nechipruk AV PLoS Genet. 2013). These observations are exciting but need an extensive, thorough and more rigorous analysis. We thus believe that it is better to investigate these preliminary observations as an independent research study that involves identifying how RUFY3 binds to kinesin and dynein-dynactin machinery and whether RUFY3 specifically activates dynein-dynactin complex but does not activate Kinesin-1.

[Redacted]

Specific points that require revisions:

1. In Figure 4C, the amount of RUFY3 co-IPing with JIP4 is very little. Quantification is necessary to determine whether this is above negative control levels. In addition, some explanation is needed for why this interaction is much weaker from the reverse co-IP of RUFY3 in figure 4B.

Response: We have included the quantification for these experiments (now **Fig. 5C-F** in the revised manuscript). We consistently see less co-IP of endogenous RUFY3 with JIP4 than co-IP of endogenous JIP4 with RUFY3. One possible explanation for these results could be related to different expression levels of RUFY3 and JIP4 in the cells with JIP4 (<https://www.proteinatlas.org/ENSG00000008294-SPAG9/cell+line>) being expressed at much higher levels compared to RUFY3 (<https://www.proteinatlas.org/ENSG00000018189-RUFY3/cell+line>). Another potential explanation that we cannot rule out at this stage is that anti-JIP4 antibody is interfering with RUFY3 binding.

2. The Co-IP in Figure 4F suggests that Arl8 is required for the RUFY3-JIP4 interaction, but clearly RUFY3 and JIP4 co-IP at native levels of Arl8 as shown in Figure 4B. The correct experiment here is to perform a quantitative blot and compare the amounts of JIP4-RUFY3 association from steady state lysates and lysates of cells expressing Arl8-HA. Quantification should demonstrate the increase in the interaction in cells that express Arl8.

Response: We think this is a misunderstanding, as in this experiment we had performed direct IP of Arl8b-HA to check for co-IP of JIP4 upon increasing the levels of RUFY3 in the cells. Thus, the original experiment (Fig. 4F) shows that RUFY3 is a linker between Arl8b and JIP4. We have now added the quantification of this experiment, which shows an increase in the JIP4-Arl8b association upon RUFY3- FLAG expression (**Fig. 5J-K**). Further, we have quantified the confocal experiments corroborating the same findings (**Fig. 5G-I**). These experiments suggest that RUFY3 acts as a linker between Arl8b and JIP4.

3. In Figure 5F, the decrease in the levels of DIC in the lysosomal fractions are not convincing. The amounts look comparable across the fractions of 1-3. Quantifications are required. Levels of DIC should be normalized to LAMP1 intensity and derived as percentage of total of DIC band intensities across the entire gradient for the individual fractions or for the fractions 1-3 or 1-4 combined. Clearly there is a shift in the knock-down with more lysosomes found in fraction 4, so this should be accounted in the quantifications. In addition, the Arl8 blot in this panel is not publication-quality.

Response: We thank both reviewers #2 and #3 for the similar suggestion. We have now quantified DIC levels in fractions 1-4 combined in control and RUFY3-depleted cells (**Fig. 6G**). We see a decrease in DIC levels in lysosomal fractions upon RUFY3 silencing in all the replicate experiments. Further, we have replaced the blot image with a better representative image for all the markers (**Fig. 6F**).

4. Figure 5G: The GFP fluorescence is very low and not visible in these panels. GFP-RILP and -TMEM55B appear as smudges, and unclear if they localize to lysosomes. This needs to be improved.

Response: We thank the reviewer for the suggestion. We have replaced the image with a better representative image and included the panel showing the GFP signal for clarity (**Fig. 6H-J**).

5. Figure 6. The quantifications that report changes in lysosome size are difficult to trust as reliable. Particularly, in the case of the rescue experiment. The perinuclear clustering of lysosomes makes it nearly impossible to distinguish individual lysosomes in the perinuclear area in order to accurately measure their size. This is a major caveat and it's unclear how the authors were able to derive faithful quantifications on lysosome size from perinuclear areas.

Response: We agree with the reviewer comment, and thus we have removed lysosome size quantification from the figure and text. We had already measured lysosome diameter by EM (**Fig. 7G-J**) and SIM (**Fig. 7K-N**), which are far better resolved.

6. The relationship between tubulation and fission events is unclear, and it's hard to follow the logic and data. Perhaps, it would be best to quantify tubulation events separately from fission events. There is no demonstration of actual fission in figure - a lysosome being split into two daughter lysosomes. Also, the presumption is that kinesin is mediating the tubulation of lysosomes by pulling membrane along microtubules, but why wouldn't dynein perform the same function? The authors should be also aware of data implicating JIP4 in actin-mediated (WASH) mechanisms of endosomal tubulation. The authors should make an effort to improve conceptually this part of the manuscript. This is a bit weak as it's also unclear how the tubulation/fission relates to the acidification aspects. The latter is part of the lysosome maturation process, and the data imply that retrograde movement of lysosomes is required for lysosome maturation in both size and acidification. But this is not clearly communicated; if indeed this is what authors think it's happening.

Response: Based on the suggestions from reviewers #1 and #2, we tried different strategies to observe events by live-cell imaging that show a distinct tubulation and subsequent fission of the late endocytic compartments. We employed Arl8b- GFP/GFP-Rab7/LAMP1-GFP along with SiR-Lysosome/Lysotracker in these experiments. Under none of these conditions, we obtained data that convincingly shows clear tubulation and then fission. This issue could be because lysosome tubulation and fission events are challenging to capture in HeLa cells than autolysosome reformation events where longer tubulation is observed. Indeed, we captured videos showing LAMP1 tubulation when we starved HeLa cells for 8 hr (**Fig. B**). Thus, we have chosen to tone down our claim of how lysosome size is reduced upon RUFY3 depletion and focus our revised study on clarifying the role of RUFY3 as a specific dynein adaptor for Arl8b-positive endosomes.

It is unclear how the selective autophagy data (i.e., puromycin-induced aggregates) relate to the RUFY3- mediated retrograde movement of lysosomes. This part of the manuscript seems to be stretching far and beyond the main focus. I recommend to remove it as it does not appear to be directly relevant, and in its place, address some obvious questions about Arl8 and RUFY3 with respect to kinesin interaction and Rab7 GTPase. It is critical to provide some clarity on RUFY3 with respect to known

interactions of its binding partners, and whether these are mutually exclusive or all in the same complex (see above).

Response: As commented by reviewers #1 and #2, we also agree that selective autophagy is farther from the focus of this work. Thus, we have removed the Puromycin-induced p62/Ub aggregate data in this revised version of the manuscript.

Minor Comment:

- Please, add kD next to every number that corresponds to molecular weight (MW) markers in gel data and add the indication "MW" on top

Response: We have added this information in the revised figures.

Response to Reviewer #3 Comments:

Kumar et al. present a study on RUFY3, a protein identified as an Arl8B interactor. They identify two residues in RUFY3 required for Arl8B binding and find that co-expression of RUFY3 with Arl8B drives lysosomes to the perinuclear region. Via immunoprecipitation-mass spectrometry of RUFY3, they identify JIP4, dynein, and dynactin components as associated with RUFY3, and show that RUFY3, JIP4, and dynein are required for perinuclear distribution of lysosomes and regulate lysosome size. Together, their data suggests a model by which RUFY3 recruits dynein to lysosomes via an interaction with JIP4. Overall, this is a very thorough, well-written paper identifying the novel adaptor RUFY3 and its mode of linking dynein to lysosomes. In most cases, the work strongly supports the claims made and makes a substantial contribution to the field. Overall, I recommend publication with the following revisions:

Response: We appreciate the positive feedback from the reviewer on our study.

Major edits:

Figure 1F is a bit hard to follow. Figure 1F should have labels clearly defining which GST-RUFY3 protein is conjugated to beads in each lane, in addition to the arrows denoting size to the right. The ponceau stain gel is also hard to see. The WT GST- RUFY3 prep appears to have pretty low expression and isn't super clean. Ideally, the authors would optimize expression of GST-RUFY3 and clean up the prep. However, this data combined with their other data makes a pretty compelling case for a direct interaction, so at the very least, the authors should perform a western blot to show alongside the Ponceau stain gel with a probe for GST or RUFY3 to ensure that RUFY3 is being expressed and to identify which band is RUFY3 in the WT condition (and potentially determine whether the other bands are degradation products). Finally, I couldn't find details on conditions for the GST-RUFY3 pulldown of His-Arl8B in the methods, so these should be added.

Response: To address this point, we have replaced the figure with a better Ponceau stained membrane (**Fig. 1F**). As suggested by the reviewer, we have probed the membrane using an anti-GST antibody but owing to the space limitation in **Fig. 1**, we have included this data in **Supplementary Fig. S1F**. Further, we apologies for the oversight of not including the protocol for performing the direct protein-protein interaction

assay for GST-RUFY3 with His-Arl8b. This information is added in the material and method section of the revised manuscript (lines #1326-1336).

The findings in Figure 5F do not appear to support the claim made that DIC level is reduced in lysosomal fractions following RUFY3 depletion. To my eye, DIC levels do not appear to be reduced in the lysosomal fractions (highest LAMP1 signal in fractions ~1-4), but appear to be reduced in later fractions. This finding is also a bit hard to follow as DIC association does not appear to correspond with LAMP1 presence to begin with, making a further "loss" hard to identify. Clarification of this assay, better labeling and description, and quantification of multiple replicates is needed.

Response: We thank the reviewer for the suggestion. We have now provided a better representative image for this experiment with more detailed labeling (Fig. 6F) and added quantification of the data from three independent experiments (Fig. 6G). As shown in Fig. 6F, DIC levels were reduced in LAMP1 fractions (#1-4) upon RUFY3 depletion. As noted by the reviewer, dynein is present in almost all the fractions and not just the LAMP1 fraction, which could be because of dynein association with other organelle membranes such as mitochondria.

Does RUFY3 directly bind JIP4? Or are there other suspected intermediates between these proteins? In the discussion, the authors claim that RUFY3 "joins the league of other late endosomal/lysosomal proteins, including RILP, TRPML1, TMEM55B and SEPT9, which interact with dynein-dynactin retrograde motor either directly or via binding to dynein adaptors JIP3 or JIP4", but they never show that RUFY3 directly binds JIP4. I suggest that the authors perform this experiment with recombinant proteins or be clearer on this point in the results and discussion. Is kinesin-1 still present on the perinuclear lysosomes containing RUFY3 and dynein? Knowing that could get at the mechanism of Arl8b-mediated dynein/kinesin competition. Are both dynein and kinesin present and associated with Arl8b on the lysosomes and one wins out depending on the circumstance? Or are they recruited/kicked off depending on context/presence or absence of RUFY3? The authors could either perform this experiment or postulate further in the discussion.

Response: We attempted to purify recombinant full-length JIP4 from *E. coli*; however, the recombinant protein was not soluble and was present in the pellet fraction post-sonication. Therefore, we tried immunopurification of FLAG-tagged-JIP4 from mammalian cells (HEK293T) using anti-FLAG resin and eluted the protein by competitively binding the beads to FLAG peptide. As shown in Fig. 5A-B, this "semi-purified" JIP4 was pulldown specifically with GST-RUFY3 but not GST-only control. Thus, we could not establish whether JIP4 directly binds to RUFY3, but our data strongly suggests a direct interaction between RUFY3 and JIP4.

Reviewer 2 also asked a similar question on the RUFY3 relationship to the Arl8b-SKIP-Kinesin-1 complex. Our preliminary findings show that RUFY3 distinctly promotes SKIP-Kinesin-1 interaction with Arl8b (Fig. D-F). Based on this data, we hypothesize that RUFY3 promotes recruitment of kinesin on Arl8b-positive compartments but that this kinesin is auto-inhibited and unable to mediate peripheral motility of these compartments. Increasing expression of PLEKHM2/SKIP relieves kinesin autoinhibition, as observed that lysosomes move towards the cell periphery in SKIP overexpressing cells (Fig. H). Thus, SKIP overcomes RUFY3's effect on lysosome positioning and moves the lysosome to the periphery. We think these observations hint towards the role

of RUFY3 as an adaptor that can bind to both dynein and kinesin, and probably RUFY3 promotes loading of inactive kinesin on lysosomes. Arl8b-binding to SKIP and its subsequent recruitment then activates kinesin-1.

Although these observations are exciting, they need an extensive, thorough and more rigorous analysis. We thus believe that it is better to investigate these preliminary observations as an independent research study that involves identifying how RUFY3 binds to kinesin and dynein-dynactin machinery. As suggested by the reviewer, we have improved our manuscript by discussing the RUFY3 relationship to the Arl8b-SKIP- kinesin-1 complex (**lines #613-630**).

Minor edits:

In all cases where asterisks are used to mark transfected cells, it would be useful to also include asterisks or another denotation in the merged images.

Response: We have now added asterisks in the merged images.

In Figure 7I and J, it would be useful to have arrows denoting fission events.

Response: These figures are now removed from the revised manuscript.

The paragraph concerning Figure 8K-M is a bit hard to read. It seems the idea for Figure 8K-M is that in RUFY3-depleted cells, there is a delay in the timing of the normal aggregate formation and clearance process, but this idea isn't super formulated until the last sentence of the paragraph. It might be useful to have a sentence describing the standard timing of the aggregation formation/clearance process at the beginning of this paragraph, and then use that as a reference to describe the different defects seen in RUFY3-depleted cells.

Response: In agreement with all the reviewers, we have now removed these figures from the revised manuscript.

Page 12: "drives lysosomes accumulation near the plasma membrane (see inset, Fig. 2F)".  "drives lysosome accumulation" or "drives accumulation of lysosomes"

Response: The text has been modified (lines #250-252).

Page 16:

"These conclusions led to a hypothesis that RUFY3 recruits dynein motor on lysosomes and thereby mediate dynein-dependent lysosomal perinuclear positioning." 

"These conclusions led to a hypothesis that RUFY3 recruits the dynein motor on lysosomes and thereby mediates dynein-dependent lysosomal perinuclear positioning."

Response: The text has been modified (lines #398-400).

Page 17: "Previous studies have shown that the perinuclear and the peripheral pools of lysosomes have few differential characteristics and functions wherein the peripheral pool of lysosomes is more poised for crosstalk and fusion with the plasma membrane and serum-dependent-mTORC1 activation (Jia and Bonifacino, 2019; Korolchuk et al., 2011; Pu et al., 2017)." 

"Previous studies have shown that the perinuclear and the peripheral pools of lysosomes have few differential characteristics and functions. The peripheral pool of lysosomes is more poised for crosstalk and fusion with the plasma membrane and serum-dependent-mTORC1 activation (Jia and Bonifacino, 2019; Korolchuk et al., 2011; Pu et al., 2017)."

Response: The text has been modified (lines #425-429).

Page 22: "This was in contrast to the control siRNA treated cells, where as expected, lysosomes were accumulated in the perinuclear region in all three conditions of starvation"  I think precise wording here is important as many RUFY3 depleted cells appear to have some lysosomal accumulation in both the perinuclear and peripheral region. I would suggest slightly modifying this sentence to "This was in contrast to the control siRNA treated cells, where as expected, lysosomes were accumulated in the perinuclear region and generally absent from the periphery in all three conditions of starvation" as both seem to be true, and the peripheral localization appears to be the most stark when looking at the images.

Response: The text has been modified (lines #493-496).

Page 24: "In contrast, we found a striking decrease in the formation of these aggregated punctae upon RUFY3 depletion (lower panel, Fig. 8K; quantification is shown in Fig. 8M)." Minor, but to me this is one of the least striking results of the whole paper. I suggest toning down the language a bit.

Response: In agreement with all the reviewers, we have removed this data from the revised manuscript.

Page 27: "A third reason could be that while markers like LAMP1 are common, but different dynein adaptors are essentially required for the motility of distinct compartments (Fig. 9A (III))."  "A third reason could be that while markers like LAMP1 are common, different dynein adaptors are essentially required for the motility of distinct compartments (Fig. 9A (III))."

Response: The text has been modified (lines #607-609).

Should cite recent Cason et al. 2021 paper "Sequential dynein effectors regulate axonal autophagosome motility in maturation-dependent pathway".

Response: We thank the reviewer for the suggestion. We have cited this publication in the revised manuscript text (line #604).

Response to Reviewer #4 Comments:

I was asked to review the mass spectrometry portion of this paper. Unfortunately, the information provided for this portion of the work is incomplete. Absolutely no details were provided by the authors on any aspect of the mass spectrometry analysis or the analysis of the mass spectrometry data (database searching). It is also standard practice to upload raw data files to a public repository such as ProteomeXchange. Until the authors provide the same level of detail for the mass spectrometry portion of this work as they do for other experiments, this work is not suitable for publication. The authors are urged to

consult with the proteomics core that ran their samples in order to get the necessary details.

Response: We sincerely apologies to the reviewer for not providing the raw data files of mass spectrometry results with our initial manuscript submission. We are not a proteomics lab, and therefore we were not aware of these practices. But this is our lack of information, and we again apologies for this oversight.

We used mass spectrometry as a tool for identifying potential interacting partners for RUFY3. Briefly, recombinant GST-RUFY3 and GST-only (as a control) proteins were used as bait proteins and incubated with lysates prepared from HEK293T cells. The coomassie stained protein bands that were specifically present in the GST-RUFY3 sample lane (Sample A: ~250-280 kDa band; Sample B: ~200-250 kDa; Sample C: ~80 kDa and Sample D: ~16 kDa band) were cut out and submitted to Taplin Mass Spectrometry Facility, Harvard Medical School (Boston) for protein identification. As a control, we cut out the whole GST-only sample lane to identify proteins that might be binding to GST protein only.

We have now revised **Supplementary Table S1** to include the complete list of proteins identified (along with peptide numbers and annotation) in samples A-D and GST-only control sample. Further, as mentioned by the reviewer, we have uploaded the RAW data of mass spectrometry that we received from the Taplin Mass Spectrometry Facility on ProteomeXchange. This information is added in the materials & methods section of the revised manuscript (**lines #1340-1353**). The submitted data are available with identifier PXD027010 and can be accessed with the following login ID and password.

Username:
reviewer_pxd027010@
ebi.ac.uk Password:
U9jysTzo

We also want to mention that we confirmed the interaction of RUFY3 with JIP4 and subunits of dynein-dynactin complex that we got as potential hits in our mass spectrometry results, as shown in **Fig. 5** using co-immunoprecipitation and GST- pulldown assays.

REVIEWERS' COMMENTS

Reviewer #1 (Remarks to the Author):

The authors have satisfactorily responded to all the comments and suggestions with extensive experimentation and analysis. I am happy to recommend publication of this manuscript. Congratulations on a very nice piece of work!

Reviewer #2 (Remarks to the Author):

The reviewers have addressed my concerns and comments. The manuscript is suitable for publication.

Reviewer #3 (Remarks to the Author):

I am satisfied with the changes that the reviewers have made and suggest the manuscript be accepted.

Reviewer #4 (Remarks to the Author):

The authors have addressed my concerns to the best of their abilities by generating Supplementary Tables with lists of identified proteins and uploading the mass spec data to ProteomeXchange. The detail in this upload is not exactly as it needs to be, but much of the detail that is missing would've needed to come from the mass spec facility at Harvard. Thus, I believe the authors made a genuine effort to address this point and I thank them for that.

This is a very interesting and informative project. The authors are to be commended for their efforts.

A very minor grammatical point:

Line 200: "little to none" should be "little to no"

Point-by-point response to reviewer's comments:

Reviewer #1 Comment:

The authors have satisfactorily responded to all the comments and suggestions with extensive experimentation and analysis. I am happy to recommend publication of this manuscript. Congratulations on a very nice piece of work!

Response: We thank the reviewer for appreciating our research work and for providing critical feedback that have allowed us to improve this study.

Reviewer #2 Comment:

The reviewers have addressed my concerns and comments. The manuscript is suitable for publication.

Response: We are grateful to the reviewer for appreciating our study.

Reviewer #3 Comment:

I am satisfied with the changes that the reviewers have made and suggest the manuscript be accepted.

Response: We appreciate the positive feedback from the reviewer on our study.

Reviewer #4 Comment:

The authors have addressed my concerns to the best of their abilities by generating Supplementary Tables with lists of identified proteins and uploading the mass spec data to ProteomeXchange. The detail in this upload is not exactly as it needs to be, but much of the detail that is missing would've needed to come from the mass spec facility at Harvard. Thus, I believe the authors made a genuine effort to address this point and I thank them for that.

This is a very interesting and informative project. The authors are to be commended for their efforts.

A very minor grammatical point:
Line 200: "little to none" should be "little to no"

Response: We would like to express our gratitude to the reviewer for appreciating our study and offering constructive comments that has helped strengthen our study.

The grammatical error is now corrected in the revised manuscript text file (line #181 and highlighted in yellow).